



# Stochastic generation of multi-site daily precipitation for the assessment of extreme floods in Switzerland

Guillaume Evin[1], Anne-Catherine Favre[1], and Benoit Hingray[1]

[1]Univ. Grenoble Alpes, CNRS, IRD, Grenoble INP*, IGE, F-38000 Grenoble, France

*Correspondence to:* Guillaume Evin (guillaume.evin@irstea.fr)

**Abstract.** Many multi-site stochastic models have been proposed for the generation of daily precipitation, but they generally focus on the reproduction of low to high precipitation events. In this paper, a multi-site daily precipitation model is proposed and aims at reproducing the statistical features of extremely rare events at different temporal and spatial scales. Recent advances and various statistical methods (regionalization, disaggregation) are considered in order to obtain a robust and appropriate

representation of the most extreme precipitation fields. Performances are shown at different temporal and spatial scales on a large region located in Switzerland.

## 1 Introduction

Stochastic precipitation generators are useful tools in risk assessment studies, the observed series of streamflows being too short to estimate the return level of very rare flooding events (e.g. decamillennial events). Typically, extreme hydrological events can

be reproduced using long series of simulated precipitations as inputs of conceptual hydrological models (Lamb et al., 2016).

In the last two decades, a fair number of precipitation models have been proposed to deal with the temporal and spatial properties of daily precipitation, for both intermittency and amount, which all have different strengths and limitations. An important proportion of these models use exogenous variables to predict statistical properties of precipitation, using generalized linear models (Chandler and Wheater, 2002; Mezghani and Hingray, 2009; Serinaldi and Kilsby, 2014b), atmospheric analogs

(Lafaysse et al., 2014) or modified Markov models (Mehrotra and Sharma, 2010). Introducing a link between exogenous atmospheric variables can be interesting to reconstruct past events, for predictions, or to downscale GCM-based simulations of future climate. Such models are classically referred to as statistical downscaling models (see Maraun et al., 2010, for a review). Closely related to this approach, weather 'types' or 'regimes' (see Ailliot et al., 2015, for a review) specifically account for different atmospheric circulation patterns. Using Hidden Markov Model with transitions between these weather

states, stochastic weather generators can then simulate various aspects of the precipitation process (Rayner et al., 2016).

Alternatively, purely stochastic precipitation models can be broadly classified into three main types:

– **Resampling methods:** The stochastic generation of precipitation fields can be performed using resampling techniques such as the K-nearest neighbors (Buishand, 1991; Yates et al., 2003). Un-observed precipitation amounts can be obtained using perturbation techniques (Sharif and Burn, 2007).

---

*Institute of Engineering Univ. Grenoble Alpes



- **Random fields:** Spatio-temporal precipitation models can simulate precipitation fields over a regular grid. These developments are particularly interesting for hydrological applications, since areal precipitation values over a catchment are directly obtained. Poisson cluster-based models (Burton et al., 2008, 2010; McRobie et al., 2013; Leonard et al., 2008) randomly simulate rain disk cells, with random centers, radius and intensity, over the study area. Meta-Gaussian models (Vischel et al., 2009; Kleiber et al., 2012; Allard and Bourotte, 2015; Baxevani and Lennartsson, 2015; Bennett et al., 2017) are based on truncated and transformed random Gaussian fields. Closely related, the turning band method can be used to simulate intermittent precipitation fields with different type of advections (Leblois and Creutin, 2013). These model structures are appealing since they are able to simulate realistic precipitation fields at fine spatial scales. However, their complexity leads to numerous technical issues during parameter estimation and simulation, notably in terms of computational cost. Moreover, they are usually unable to represent large regions, with very distinct precipitation regimes.

- **Multi-site models:** Since the pioneering work of Wilks (1998), numerous weather generators have been developed to fit directly the statistical properties of precipitation at a limited number of stations (Bárdossy and Pegram, 2009; Srikanthan and Pegram, 2009; Baigorria and Jones, 2010; Rasmussen, 2013; Chen et al., 2014; Keller et al., 2015). For both precipitation occurrence and amount, multi-site generators are able to preserve the inter-dependency between all pairs of stations, even when the area under study exhibits very different precipitation regimes (e.g. in mountainous areas).

The context of this work is the risk assessment of extreme flooding events using the "continuous simulation" method. Very long series of daily precipitation (e.g. 10,000 years) are generated for the present climate and used as inputs of a conceptual hydrological model. As the hydrological process and extreme floods are influenced by the hydrological configurations (for example, different levels of soil saturation), these precipitation scenarios must reproduce low to very extreme daily precipitation events at different temporal and spatial scales.

In this paper, we develop a precipitation model, called GWEX, which will be used to generate these long scenarios over a large area with various precipitation regimes. GWEX is applied to 105 stations of the Aare river catchment in Switzerland. As multi-site models have a flexible structure which can be applied to a large number of stations with very different precipitation regimes, GWEX relies on the structure proposed by Wilks (1998). The underlying idea is to separate the process representing the precipitation occurrences at the different stations from the process generating the amounts of the precipitation events.

In this work, we take advantage of recent studies on precipitation extremes. Papalexiou et al. (2013) and Serinaldi and Kilsby (2014a) assess the distributional behavior of 15,029 long daily precipitation records ($> 50$ years) from around the world. They conclude that heavy-tailed distributions are generally in better agreement with the observed precipitation extremes. Follow-up studies (Papalexiou and Koutsoyiannis, 2013; Serinaldi and Kilsby, 2014a) apply the extreme value theory to annual maxima and "peaks over threshold" (POTs) of a large sub-set of these records and confirm that extreme daily precipitations are not adequately represented by light-tailed distributions. Using statistical tests on 90,000 station records of daily precipitation, Cavanaugh et al. (2015) also come to the same conclusions. These findings have important implications for precipitation models:





- Light-tailed distributions, such as exponential, Gamma or Weibull distributions, which are applied in the vast majority of the existing precipitation models, often lead to an under-estimation of extreme daily precipitation amounts.

- While non-parametric densities with Gaussian kernels (Mehrotra and Sharma, 2007, 2010) offer a great flexibility to fit the observed range of precipitation amounts, their tail also belong to the domain of attraction of the Gumbel distribution and suffer from the same drawbacks.

Alternatively, current statistical procedures consisting in fitting a flexible distribution to the bulk of the observations and use it for extrapolation is highly questionable, as major assumptions are usually violated, as it has been extensively discussed by Klemeš (2000a, b). Since the tail of the distribution on precipitation amounts at each station will dictate the generation of the most extreme precipitation events, important features of GWEX are:

- to apply an heavy-tailed distribution to precipitation amounts at each station (Naveau et al., 2016), following the conclusions drawn by Papalexiou et al. (2013); Serinaldi and Kilsby (2014a); Cavanaugh et al. (2015),

- to obtain robust estimates of the shape parameter of this distribution, which indicates the heaviness of the tail, using a regionalization approach, as in Evin et al. (2016).

Furthermore, following Bárdossy and Pegram (2009), GWEX also employs the copula theory to introduce a tail dependence between the precipitation amounts simulated at the different stations.

The global methodology is first described in Section 2, with a presentation of the study area, the features of different multi-site precipitation models, and the evaluation framework, which aims at assessing the performances of GWEX at different spatial and temporal scales. Section 3 details the applications of these daily precipitation models to 105 stations located in the Aare river catchment. Section 4 synthesizes the results, focusing on the reproduction of extreme events. Section 5 concludes.

## 2 Material and methods

### 2.1 Data and study area

The Aare River basin covers the northern part of the Swiss Alps and has an area of 17,700 km$^2$. Basin elevations approximately range from 310 m.a.s.l. in Koblenz (entrance to Germany in the north) to 4270 m.a.s.l. at the Finsteraarhorn summit (in the south of the area). The mean annual precipitation for the basin as a whole is 1300 mm. The basin can be divided into five main sub-basins with different hydrometeorological regimes highly governed by regional terrain features (Jura mountains in the North-West; Northern Alps in its southern part, lowlands in the middle).

Figure 1 shows the location of the 105 precipitation stations used for the development and the evaluation of weather generators. Located within or close to the Aare river Basin, they correspond to the stations for which long daily time series of observations with less than 3 years of missing data are available during the period 1930-2014. The 105 precipitation stations cover relatively well the Aare river catchment.



The weather scenarios are used to simulate, via a conceptual hydrological model, flood scenarios for the whole Aare River Basin and for its different sub-basins. Therefore, the properties of the weather scenarios must be evaluated at different spatial and temporal scales, from high resolutions required for simulating the hydrological behavior of the system (e.g. sub-daily, 100 km$^2$) to lower resolutions relevant at the scale of the entire basin (e.g. n-days, 17,700 km$^2$).

5    Following Mezghani and Hingray (2009), a multi-scale evaluation in space and time is thus carried out. For instance, the performance of GWEX are evaluated at the station scale, at the scale of 5 and 15 sub-basins partitioning the Aare river catchment (see Figure 1), and at the scale of the entire study area (see Section 4). Note that for those evaluations, areal estimates of precipitation are obtained from the precipitation amounts at the stations using the Thiessen polygon method.

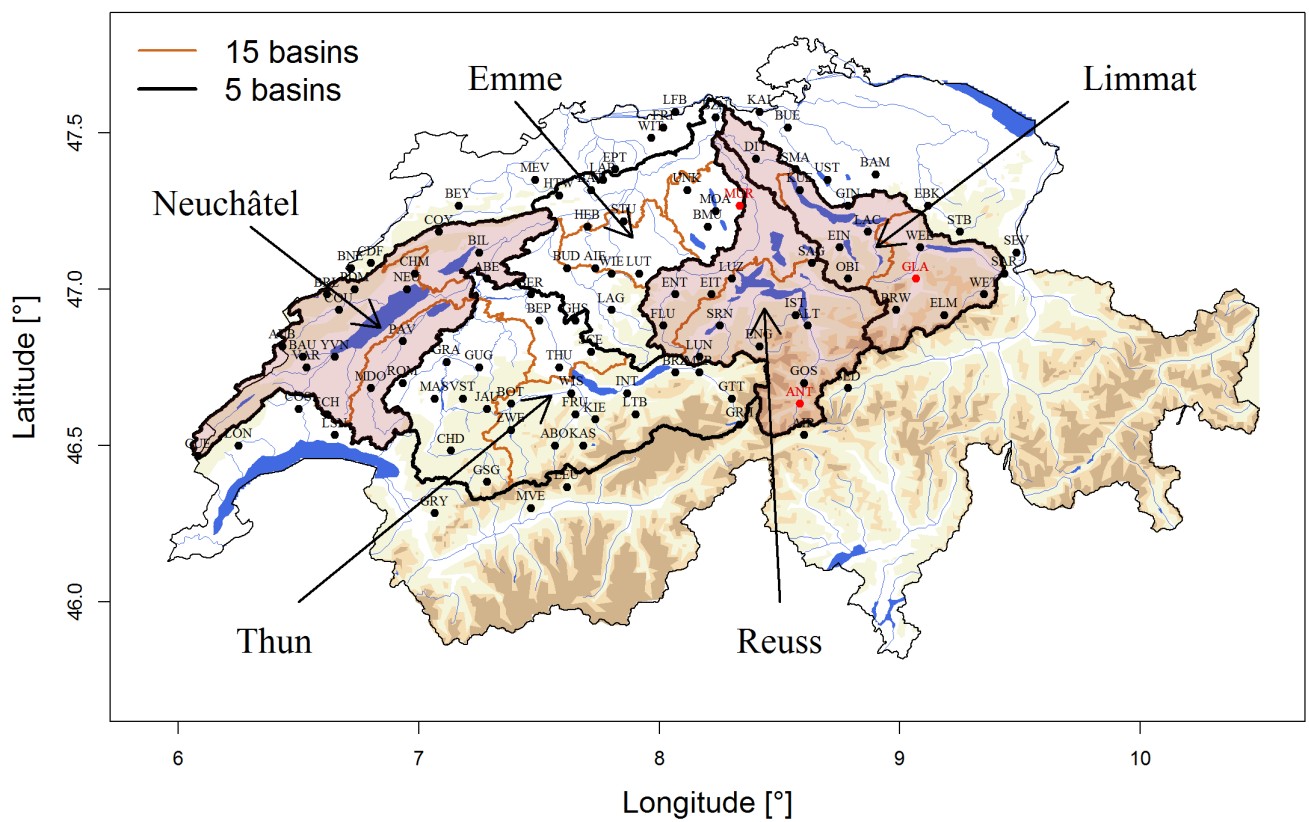

**Figure 1.** Location of the 105 precipitation stations in Switzerland. Different partitions of the Aare river catchment are considered and the names of the five main sub-basins are indicated.

## 2.2    Multi-site precipitation model

10    As indicated above, GWEX is a multi-site precipitation model which relies strongly on the structure proposed by Wilks (1998). Precipitation amounts are modeled independently of precipitation occurrences, which act as a mask.




### 2.2.1 Precipitation occurrence process

As proposed by Wilks (1998), the occurrence process can be represented by a two-state Markov chain, representing 'dry' and 'wet' days:

$$
X_t(k) = \begin{cases} 0, & \text{if day } t \text{ is dry at location } k. \\ 1, & \text{if day } t \text{ is wet at location } k. \end{cases} \tag{1}
$$

In practice, these states are obtained using a low precipitation threshold (0.2 mm). In the present case, the seasonality of the occurrence process is taken into account by estimating model parameters on a monthly basis.

**At-site occurrence process**

At each location, the temporal persistence of dry and wet events is introduced with a $p$-order Markov chain model for $X_t(k)$, which means that the probability of having a wet day at time $t$ depends only on the $p$ previous states, for days $t-1, \ldots, t-p$.
While many authors suppose that a first-order Markov is sufficient (e.g. Wilks, 1998; Keller et al., 2015), Srikanthan and Pegram (2009) apply a 4-order Markov chain and show that it improves the reproduction of dry/wet period lengths. In this study, different orders for this Markov chain are considered. At each site, the probability of having a wet day at day $t$ is given by the transition probability $\Pr\{X_t(k) = 1 | X_{t-1}(k), \ldots, X_{t-p}(k)\}$. This Markov chain is thus fully characterized by a transition matrix $\Pi$ with dimensions $2^p$.
These transition probabilities are estimated directly by the proportion of wet days $X_t(k) = 1$ following observed sequences $\{X_{t-1}(k), \ldots, X_{t-p}(k)\}$.

**Spatial occurrence process**

Let $\rho = \text{Corr}(X_t(k), X_t(l))$ denote the inter-site correlation between the states $X_t(k)$ and $X_t(l)$. Following Srikanthan and Pegram (2009), $\rho$ can be expressed as:

$$
\quad \rho = \frac{\pi_{00}(k,l) - \pi_0(k)\pi_0(l)}{\sqrt{\pi_0(k)\pi_1(k)}\sqrt{\pi_0(l)\pi_1(l)}}, \tag{2}
$$

where $\pi_0(s) = \Pr\{X_t(s) = 0\}$ and $\pi_1(s) = \Pr\{X_t(s) = 1\}$ denote the probabilities of having dry and wet states at location $s$, respectively, and $\pi_{00}(k,l) = \Pr\{X_t(k) = 0, X_t(l) = 0\}$ denotes the joint probability of having dry states at both locations $k$ and $l$. $\hat{\rho}$ estimates can thus be obtained using the empirical (i.e. observed) counterparts of these probabilities.

Following Wilks (1998), for two locations $k$ and $l$, a bivariate normal distribution with mean $\mathbf{0}$, variance $\mathbf{1}$ and a correlation
parameter $\omega$ can be employed to reproduce $\hat{\rho}$. More precisely, Gaussian variates can be converted to uniform variates using the probability integral transform. The correlated uniform variates are then compared to the transition probabilities in order to generate new states $X_t(k)$ and $X_t(l)$ (see Wilks, 1998, for further details).





The relationship between $\omega$ and $\hat{\rho}$ is not direct since the at-site occurrence process also influences $\hat{\rho}$ (Wilks, 1998). Figure 2 illustrates this relationship for two close stations, GOS and ANT. For these two stations, transition probabilities with a Markov chain of order 4 are computed for the month of January. Given these transition probabilities, stochastic simulations are then generated for different values of $\omega$, leading to different values of $\rho$. Since this relationship is monotonic (see Fig. 2), a value $\omega$

5    corresponding to an empirical estimate $\hat{\rho}$ can be found iteratively until the evaluation of the correlation between the simulated precipitation states, $\rho$, matches $\hat{\rho}$. It must be noticed that a very high value for $\hat{\rho}$ cannot always be reached, even if $\omega = 1$. This is however a situation which rarely occurs in practice.

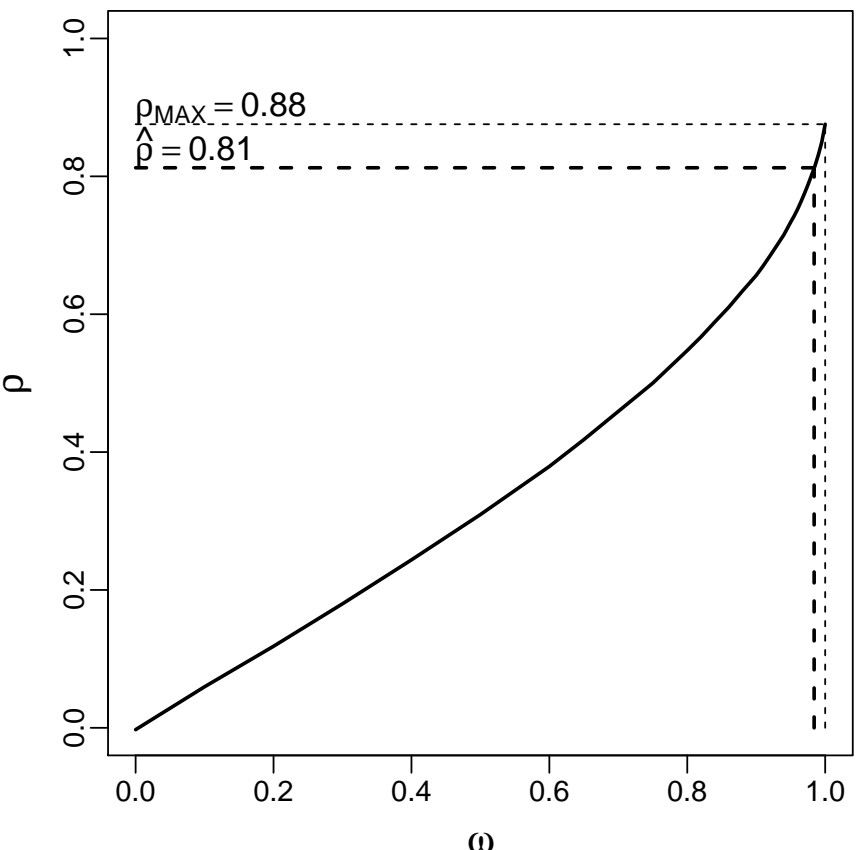

**Figure 2.** Illustration of the relationship between $\omega$ and $\rho$ for the month of January, and for stations GOS and ANT. A Markov chain of order 4 is considered in this example. The correlation between the observed states is $\hat{\rho} = 0.81$ and can be reproduced using a bivariate Gaussian distribution with a correlation parameter of $\omega = 0.98$. The maximum correlation $\rho$ which can be obtained if $\omega = 1$ is $\rho_{MAX} = 0.88$.

The cross-correlations $\omega$ are estimated for all possible pairs of stations and the corresponding correlation matrix is denoted by $\Omega_X$. If $\Omega_X$ is not positive-definite, the closest positive-definite matrix is considered (Rousseeuw and Molenberghs, 1993;

10    Rebonato and Jaeckel, 2011).





### 2.2.2 Precipitation intensity process

Given the occurrence of precipitation $X_t(k)$ at different locations $k$, GWEX generates the amounts of precipitation $Y_t(k)$ using:

- marginal heavy-tail distributions,

– a tail-dependent spatial distribution,

- an autocorrelated temporal process.

Similarly to the occurrence process, the seasonal aspect of the precipitation intensity is taken into account by performing the parameter estimation for each month, on a 3-month moving window.

**Marginal distributions**

At a given location $k$, daily precipitations have often been modeled by light-tailed distributions: exponential and Weibull distributions (Bárdossy and Pegram, 2009); gamma distributions (Srikanthan and Pegram, 2009; Mezghani and Hingray, 2009); mixture of exponential distributions (Wilks, 1998; Keller et al., 2015); mixture of gamma distributions (Chen et al., 2014). However, as shown by many recent studies on a very large number of daily precipitation series (Papalexiou et al., 2013; Serinaldi and Kilsby, 2014a; Cavanaugh et al., 2015), exponentially decaying tails often result in a severe underestimation of

extreme event probabilities. The introduction of an heavy-tailed distribution is thus crucial for the reproduction of the most extreme precipitation events.

In this work, the distribution representing the precipitation intensity at each location can be described by a smooth transition between a gamma-like distribution and an heavy-tailed Generalized Pareto distribution (GPD). This transition is obtained via a transformation function, $G(\nu)$, such that the whole range of precipitation intensities is modeled without a threshold selection

(Naveau et al., 2016):

$$F_Y\{Y_t(k)\} = G\Big[H_\xi\big\{Y_t(k)/\sigma\big\}\Big], \tag{3}$$

where

$$H_\xi(z) = \begin{cases} 1 - (1 + \xi z)_+^{-1/\xi} & \text{if } \xi \neq 0, \\ 1 - e^{-z} & \text{if } \xi = 0, \end{cases} \tag{4}$$

with $a_+ = \max(a, 0)$, is the standard cumulative distribution function of the GPD, $\sigma > 0$ is a scale parameter and $G(\nu) =$

$\nu^\kappa, \kappa > 0$. Thus, a 3-parameter set $\{\sigma, \kappa, \xi\}$ needs to be estimated at each station.

This distribution has been proposed by Papastathopoulos and Tawn (2013) under the name of extended GP-Type III distribution and has been shown to model adequately precipitation intensities (Naveau et al., 2016). Here, we refer to this distribution under the name of E-GPD.





Local estimations of the GPD tail exhibiting a lack of robustness, the $\xi$ parameter of the E-GPD is estimated using a regionalization method similar to Evin et al. (2016). Neighborhoods around each station are first obtained using homogeneity tests, following the concept of regions-of influence (RoI) proposed by Burn (1990). The $\xi$ parameters are then estimated using the precipitation data gathered in this region (see section 3.2 for details). The two remaining parameters, the scale parameter

$\sigma$ and the parameter of the transformation $\kappa$, are estimated at each station using a method of moments based on probability weighted moments (see Naveau et al., 2016, for further details).

**Spatial and temporal dependence of precipitation amounts**

Spatial and temporal dependence of precipitation amounts is represented using a Multivariate Autoregressive model of order 1, MAR(1). A MAR(1) process has been used by different authors (Bárdossy and Pegram, 2009; Rasmussen, 2013) to represent

simultaneously spatial and temporal dependences. Let $\mathbf{Z}_t$ denote a vector of $K$ Gaussian variates with mean 0. The MAR(1) process can be described as follows:

$$\mathbf{Z}_t = \mathbf{A}\mathbf{Z}_{t-1} + \epsilon_t, \tag{5}$$

where $\mathbf{A}$ is a $K \times K$ matrix and $\epsilon_t$ is a random $K \times 1$ noise vector. The elements of $\epsilon_t$ have zero means and are independent of the elements of $\mathbf{Z}_{t-1}$. The covariance matrix of $\epsilon_t$ is denoted by $\Omega_Z$. Following Bárdossy and Pegram (2009), $\mathbf{A}$ is taken to

be a diagonal matrix whose diagonal elements are the lag-1 serial correlation coefficients. The matrix $\Omega_Z$ is then obtained as:

$$\Omega_Z = \mathbf{M}_0 - \mathbf{A}\mathbf{M}_0'\mathbf{A}, \tag{6}$$

where $\mathbf{M}_0$ is the covariance matrix of $\mathbf{Z}_t$, which indicates the degree of spatial dependence between each pair of stations.

Elements of $\mathbf{Z}_t$ are first obtained using the following transformation:

$$Z_t(k) = \Phi^{-1}\big[\hat{F}\{Y_t(k)\}\big], \tag{7}$$

where $\hat{F}$ is the empirical distribution function and $\Phi[.]$ indicates the standard normal cumulative distribution function.

Using the method of moments, $\mathbf{M}_0$ is estimated directly by pairwise covariances between the elements of $\mathbf{Z}_t$ using the Kendall's rank correlation $\tau$, which can be directly related to the Pearson correlation coefficient $\rho_P$ for elliptical distributions (McNeil et al., 2005, p.97):

$$\rho_P = \sin\left(\frac{\pi}{2} \times \tau\right), \tag{8}$$

including Gaussian and Student multivariate distributions. The Kendall's $\tau$ does not depend on the marginal distributions, unlike the linear Pearson correlation $\rho_P$, and has the advantage to be a robust estimator of the degree of dependence, since it is





calculated from the ranks of the data alone. Since $\Omega_Z$ is not necessarily positive-definite (see Eq. 6), the closest positive-definite matrix is taken as the covariance matrix of $\epsilon_t$ if necessary.

Innovations $\epsilon_t$ are often assumed to follow a standard multivariate normal distribution, which means that their dependence structure is modeled by a Gaussian copula. However, the upper tail dependence of the Gaussian copula is 0, which means that

extreme precipitation events simulated from a Gaussian copula are not spatially dependent. This motivates the use of a Student copula to represent the dependence structure of $\epsilon_t$, for which an additional parameter, $\nu$, is related to the tail dependence. Given $\Omega_Z$, the parameter $\nu$ is estimated by maximizing the likelihood, as described in McNeil et al. (2005, Section 5.5.3.).

### 2.2.3   Model versions

Different versions of the proposed multi-site precipitation model are considered in this paper. The performances of these

different versions will then be presented in Section 4.

**Wilks**

A first benchmark version of the multi-site model, 'Wilks', is considered, which closely matches the multi-site model proposed by Wilks (1998). In particular:

- The at-site occurrence process is a Markov chain of order 1.

– A threshold of 0.2 mm separates dry and wet states.

- The marginal distribution on precipitation amounts is a mixed-exponential distribution, for which the pdf is defined as:

$$f(x) = \frac{w}{\beta_1} \exp\left(-\frac{x}{\beta_1}\right) + \frac{1-w}{\beta_2} \exp\left(-\frac{x}{\beta_2}\right). \tag{9}$$

The parameters $w$, $\beta_1$ and $\beta_2$ are estimated using the Expectation-Maximisation (EM) method (Dempster et al., 1977).

- Precipitation amounts are not considered to be temporally correlated, i.e. the matrix $\mathbf{A}$ in equation 5 is a zero matrix.

Furthermore, innovations $\epsilon_t$ follow a standard multivariate normal distribution.

**GWEX-1D**

A first version of the GWEX model presented in this section, labeled GWEX-1D, has the following specifications:

- The at-site occurrence process is a Markov chain of order 4.

- A threshold of 0.2 mm separates dry and wet states.

– The marginal distribution on precipitation amounts is the E-GPD distribution.

- Precipitation amounts follow a MAR(1) process with innovations modelled by a Student copula.





**GWEX-3D**

As will be shown in Section 4, GWEX-1D model tends to underestimate extreme amounts for different temporal scales (e.g. 3 days). It motivated the investigation of an alternative version, GWEX-3D. GWEX-3D is applied to 3-day precipitation amounts, with the same specifications than GWEX-1D, except that:

– The at-site occurrence process is a Markov chain of order 1.

– A threshold of 0.5 mm separates dry and wet states.

With GWEX-3D, daily scenarios are first generated at a 3-day scale, and disaggregated at a daily scale using a method of fragments (see, e.g., Buishand, 1991). Simulated 3-day amounts are disaggregated using the temporal structures of the closest observed 3-day amounts, in terms of similarity of the spatial fields. The details of the disaggregation method are provided in
Appendix A. Compared to GWEX-1D, GWEX-3D model presents the following advantages:

– 3-day precipitation amounts are directly modeled and have a better chance to be adequately reproduced,

– the disaggregation of 3-day precipitation amounts creates a inherent link between the occurrence and the intensity processes. For very extreme precipitation events, we can suspect that these processes are dependent (higher chance to be in a wet state over the whole Aare river catchment, as well as large and persistent precipitation amounts).

**2.3  Multi-scale evaluation**

In this study, the performances of the multi-site precipitation models are assessed using a multi-scale evaluation, temporally and spatially. We investigate if the statistical properties of precipitation data are adequately reproduced at the scale of the stations, and for different partitions of the Aare river catchment (see Figure 1). In order to achieve this, 100 daily precipitation scenarios are generated, each scenario having a length of 100 years.

For the different evaluated statistics, performances are categorized according to the comprehensive and systematic evaluation (CASE) framework proposed by Bennett et al. (2017). More precisely, one of three categories: 'good', 'fair' and 'poor' performance, is assigned to each metric according to the agreement between the observed metric and the simulated metrics computed from the 100 scenarios. Table 1 summarizes the tests leading to each performance category. 'good' performances are obtained when the observed metric is inside 90% limits of the 100 simulated metrics (case 1). It indicates that simulated
metrics are in good agreement with the observed one. However, we can obviously expect that observed metrics lie outside these limits without indicating a failure of the model. In this case, 'fair' performances are assigned, according to two different rules:

1. Case 2: The observed metric is outside 90% limits but within three standard deviations from the simulated mean, which corresponds to the 99.7% limits if we assume that the uncertainty in the statistics is normally distributed. This case covers the situation where we could expect that the observed metric is outside the 90% limits due to the sampling uncertainty.



2. Case 3: The absolute relative difference $|(S_{obs} - \bar{\mathbf{S}}_{sim})/S_{obs}|$ between the observed metric $S_{obs}$ and the mean of the simulated metrics $\bar{\mathbf{S}}_{sim}$ is 5% or less. If the variability of the simulated metrics is very small, it can happen that the observed metric lie outside the 99.7% limits without being too far from the simulated mean in terms of relative difference.

Otherwise, we consider that 'poor' performances have been obtained, which indicates that the model fails to reproduce this particular statistical property.

In summary, 'good' performances represent cases for which the observed metric is clearly well reproduced by the model, whereas 'fair' performances indicate a reasonable match between the observed and the simulated metrics. The number of metrics for which 'poor' performances are obtained is thus the first criteria indicating the overall performance of a model.

**Table 1.** Performance categorization criteria from Bennett et al. (2017).

| Performance Classification | Key | Test |
|---|---|---|
| 'good' | 🟩 | Observed metric inside 90% limits (case 1) |
| 'fair' | 🟨 | Observed metric outside 90% limits but within the 99.7% limits (case 2) OR Absolute relative difference between the observed metric and the average simulated metrics is 5% or less (case 3) |
| 'poor' | 🟥 | Otherwise (case 4) |

For illustration purposes, we also present the results of the evaluation for three precipitation stations and sub-catchments corresponding to different hydrological regimes (see table 2). Figure 1 shows the 3 (over 105) selected precipitation stations and the 3 (over 5) representative catchments. Station ANT (at Andermatt) is located in a glacial catchment, station GLA (at Glarus) in a nival catchment and station MUR (at Muri) in a pluvial catchment.

Two selected sub-catchments (Reuss and Limmat) include these stations and a third sub-catchment (Neuchâtel) covers the west part of the study area.

**Table 2.** Hydrological regimes and characteristics of extreme floods in Switzerland (Froidevaux, 2014).

| | Mean elevation [m] | Season | Triggering events |
|---|---|---|---|
| Glacial | > 1900 | summer | showers + snow melt |
| Nival | 1200 − 1900 | summer, spring | showers, long rain |
| Pluvial | < 1200 | summer | long rain |





In this work, we focus on daily and 3-day precipitation maxima, high discharge events being usually triggered by meteorological events with a duration of several days, in late summer and autumn (Froidevaux, 2014).

## 3 Application

### 3.1 Split-sampling procedure

The precipitation observations are split into two parts: (1) 45 years randomly chosen among the period 1930-2014 are used to estimate the parameters (2) the 40 remaining years are used to evaluate the performances of the models. This separation between an estimation set and a validation set is crucial to test the ability of the model to adequately represent the statistical properties of events which have not been used during the fitting procedure. In this study, the multi-scale evaluation is only applied to the validation set of 40 years.

The $\xi$ parameter of the E-GPD is estimated using all available precipitation data in Switzerland, following the regionalization method described below. This approach ensures that robust estimates are obtained for this parameter, which is crucial in our context since extreme simulated precipitation amounts are highly sensitive to the $\xi$ parameter. All the other parameters are estimated with the estimation set of 45 years, following the methodology described in section 2.2.

### 3.2 Regionalization of the $\xi$ parameter

For the different stations, the $\xi$ parameter of the E-GPD (see Eq. 4) is estimated using a regionalization method. This methodology is similar to what is proposed by Evin et al. (2016) and can be summarized as follows:

1. For each station, a neighborhood is obtained using homogeneity tests. All the stations inside this region-of influence (RoI) are then considered homogeneous up to a scale factor.

2. The $\xi$ parameters are then estimated with the maximum likelihood method using the precipitation observations from all
the stations inside the RoI.

This regionalization method has been applied to the precipitation data from 666 stations available in Switzerland, for 4 different seasons:

– **Winter:** December, January and February,

– **Spring:** March, April and May,

– **Summer:** June, July and August,

– **Autumn:** September, October and November.

In this work, the estimation of the $\xi$ parameter is bounded between 0 and 0.25. When $\xi < 0$, the E-GPD distribution has an upper bound. When $\xi > 0.25$, extremely fat tails are obtained, which usually lead to unreasonable simulated precipitations. As



shown by many recent studies (see, e.g. Serinaldi and Kilsby, 2014a), negative and high estimates of $\xi$ are usually due to the parameter uncertainty and are not realistic.

For GWEX-1D, the estimation of the $\xi$ parameter is performed at a daily scale. In order to highlight spatial patterns of $\xi$ over Switzerland, we show the maps of the interpolated parameter estimates in Figure 3. Fat tails are obtained in the South and East of the Aare river catchment, particularly during spring and summer seasons. In the south of Switzerland, a region with high estimates ($\xi \sim 0.2$), highlighted in red, is obtained for the summer and autromn seasons. These high $\xi$ estimates are coherent with the presence of strong convective storms in this mountainous region during this period of the year (Rudolph and Friedrich, 2012).

For GWEX-3D, the regionalization method has also been applied at a 3-day scale (see Figure 4). The resulting estimates are similar to the ones obtained at a daily scale. However, we can notice that the very high estimates obtained during the summer season at a daily scale are lower at a 3-day scale. This seems to confirm the interpretation of these high $\xi$ estimates, i.e. the relationship between summer convective storms and high $\xi$ estimates is not as strong at a 3-day scale, since this type of storms usually have a short duration.

## 3.3 Generation of scenarios

For each multi-site precipitation model investigated in this paper (Wilks, GWEX-1D and GWEX-3D), we generate 100 daily precipitation scenarios, each scenario having a length of 100 years. These scenarios are compared to the precipitation observed during the validation period of 40 years.





**Figure 3.** Regionalization of the $\xi$ parameter at a daily scale, for the different seasons.





Season 1: DEC, JAN, FEB

Season 2: MAR, APR, MAY

Season 3: JUN, JUL, AUG

Season 4: SEP, OCT, NOV

**Figure 4.** Regionalization of the $\xi$ parameter at a 3-day scale, for the different seasons.





## 4   Results

This section presents the results of the multi-scale evaluation framework (see Section 2) for several metrics related to the occurrence process of the precipitation events, daily amounts, monthly totals and precipitation extremes. As much as possible, synthetic assessments are provided, with several statistics being provided for all the spatial scales of interest. Illustrative
examples are shown in order to support the conclusions drawn from these synthetic results.

### 4.1   Occurrence process

The comparison of the monthly number of wet days obtained from observed and simulated precipitation data are shown in Figure 5. The average number of wet days is adequately reproduced by all models, with approximately 30% of cases with 'poor' performances. These 'poor' performances seem to occur mainly during the winter and spring seasons. The standard
deviation of the monthly number of wet days indicates the inter-annual variability of this metric. While the magnitudes of the standard deviations from the simulated precipitations roughly match the corresponding observed standard deviations, it seems that the highest observed variabilities are under-estimated by all the models, this defect being more apparent for the Wilks model.

Figures 6 and 7 show the distributions of observed and simulated dry and wet spells, respectively, for the three illustrative
stations. Concerning the distributions of dry spell lengths, GWEX-1D and GWEX-3D models both lead to adequate performances, the performances being classified as 'good' in 48% and 51% of the cases, respectively. The performance of Wilks model are slightly lower because of an imprecise reproduction of the frequency of the shortest dry spells. This difference of performances between Wilks and GWEX-1D models is explained by the order of the Markov chain used to simulate the transitions between dry and wet states, which is the only difference between the occurrence processes of these models. The 4-order
Markov of the GWEX-1D model seems to be a more adequate representation of these transitions than the first-order Markov chain of the Wilks model, confirming previous findings (Srikanthan and Pegram, 2009).

The frequencies of wet spell lengths are adequately reproduced by Wilks and GWEX-1D models, with more than 50% of 'good' performances. The lower overall performance of GWEX-3D for this metric is due to a slight underestimation of the longest wet spells for some stations (which is not the case for the stations shown in Fig. 7).





**Figure 5.** At site number of wet days for all sites and months: inter-annual mean and standard deviation (sd). 90% probability limits are shown for the different seasons. The overall performance represents a percentage of 'good', 'fair' and 'poor' performances for all sites and months ($105 \times 12 = 1260$ cases).



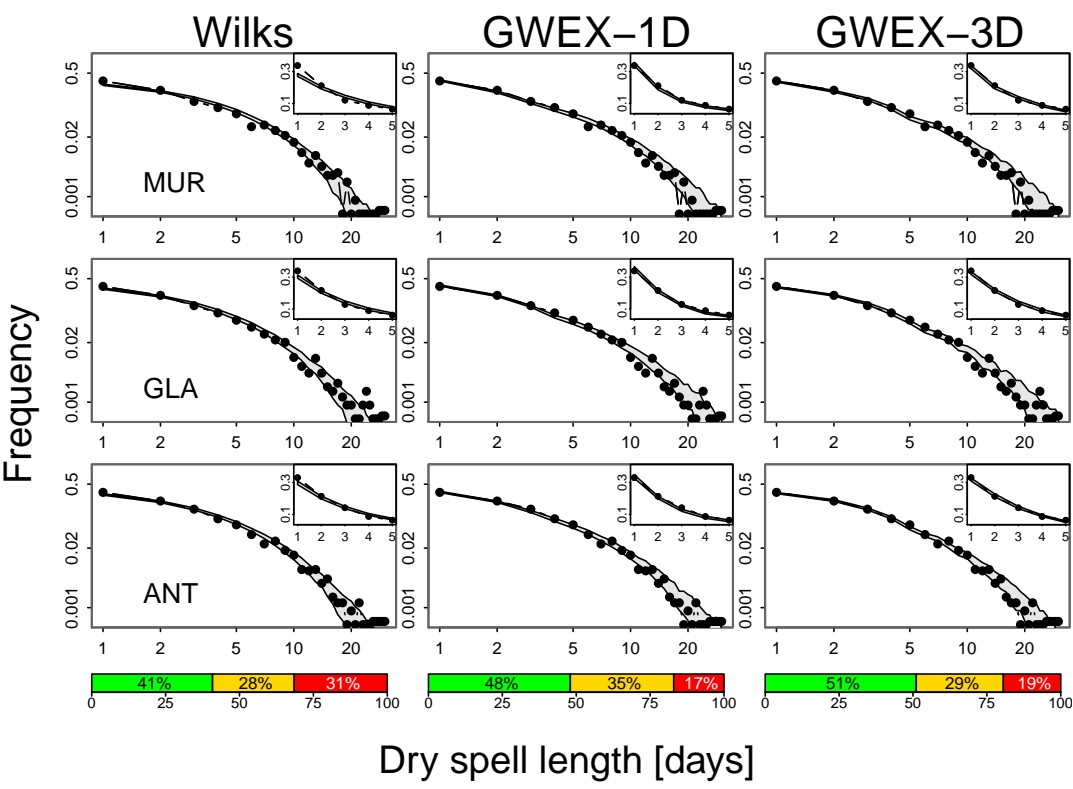

**Figure 6.** Distribution of dry spell lengths at the stations: 90% probability limits are shown. The overall performance represents a percentage of all sites. Inset plots provide a zoom for durations of 1 to 5 days.





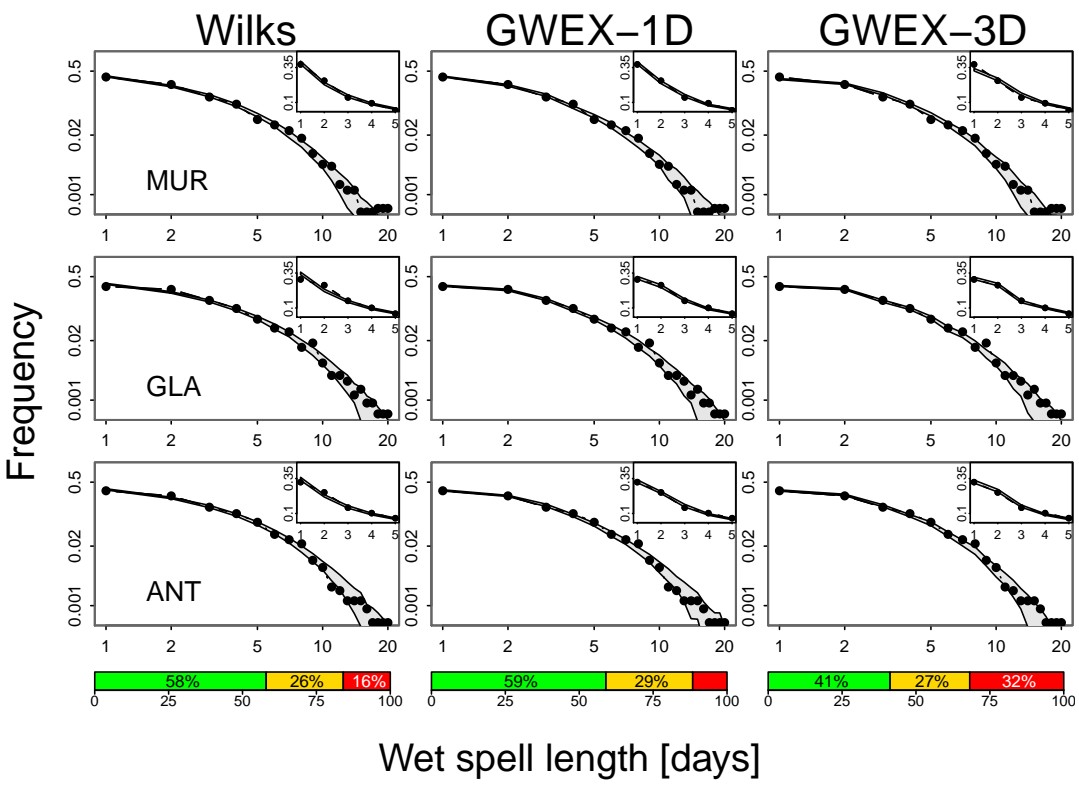

**Figure 7.** Distribution of wet spell lengths at the stations: 90% probability limits are shown. The overall performance represents a percentage of all sites. Inset plots provide a zoom for durations of 1 to 5 days.





## 4.2 Daily amounts

The reproduction of precipitation amounts at a daily scale is assessed in Figure 8, for all spatial scales and months. For all models, we obtain a reasonable agreement between observed and simulated average daily amounts (90% limits close to the 1:1 line), with more than 40% of 'good' cases and less than 30% of 'poor' cases. The standard deviations of these daily

5    amounts is also adequately reproduced (Fig. 8, bottom plots). However, we can notice that these standard deviations are slightly underestimated by models Wilks and GWEX-1D at the scale of the basins, which is not the case of the GWEX-3D model.

**Figure 8.** Daily amounts for all spatial scales and months: inter-annual mean and standard deviation (sd). 90% probability limits are shown. The overall performance represents a percentage of all spatial scales and months.



## 4.3 Inter-annual variability

The reproduction of the standard deviations of aggregated precipitation amounts at a monthly scale is used to assess the inter-annual variability (Figure 9), for all spatial scales and months. For the winter months, the standard deviations of these monthly totals are underestimated at all spatial scales (Fig. 9, top plots). We can clearly interpret this deficiency as an under-estimation of the inter-annual variability of these aggregated amounts. This deficiency has been identified in many stochastic precipitation models (see, e.g. Wilks and Wilby, 1999; Bennett et al., 2017) and different remedies have been proposed in the literature (Mehrotra and Sharma, 2007; Mehrotra et al., 2012). However, we can notice that this underestimation is moderate for GWEX-3D (32% of 'poor' cases against 79% for Wilks). Furthermore, this under-estimation of the inter-annual variability is not present for the summer months (Fig. 9, bottom plots).

**Figure 9.** Monthly totals for all spatial scales and months: inter-annual standard deviation (sd) for the winter (DJF) and summer (JJA) seasons. 90% probability limits are shown. The overall performance represents a percentage of all spatial scales and the months corresponding to the season.





### 4.4 Extreme precipitation amounts

Figures 10 and 11 show a comparison of the observed and simulated annual maximum precipitation for the three illustrative stations, at a daily and at a 3-day scale, respectively. At a daily scale, the three precipitation models exhibit different behaviors for the simulated maxima. Maxima from Wilks are linear on a Gumbel scale, which is expected as daily intensities are generated

5 from a mixture of exponential distributions. GWEX-1D, with the E-GDP distribution, generates larger extreme precipitation amounts than Wilks, but also than GWEX-3D. For example, at station ANT, the 95% quantile (upper limit of the 90% intervals) of the largest daily annual maxima obtained from the 100-year scenarios exceed 350 mm for GWEX-1D and is below 300 mm for GWEX-3D.

At a 3-day scale, larger discrepancies can be observed between the three models (Fig. 11). In particular, observed maxima

10 are strongly underestimated by Wilks at stations GLA and ANT, which is not the case (or, at least, not as clearly), for the two other models. We can thus assume that the temporal dependency introduced by the MAR(1) process (Eq. 5) leads to a better reproduction of the largest precipitation amounts cumulated on several days. As expected, GWEX-3D performs well at a 3-day scale, which justifies the strategy consisting in fitting directly 3-day amounts.

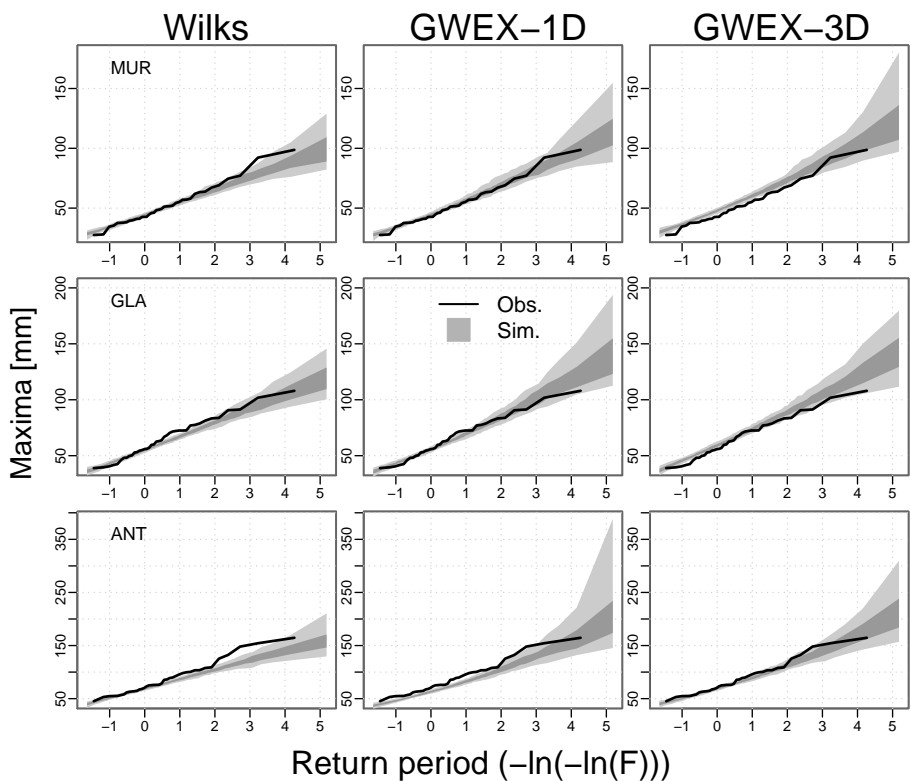

**Figure 10.** Simulated and observed daily annual maxima at the stations: 50% and 90% probability limits are shown.



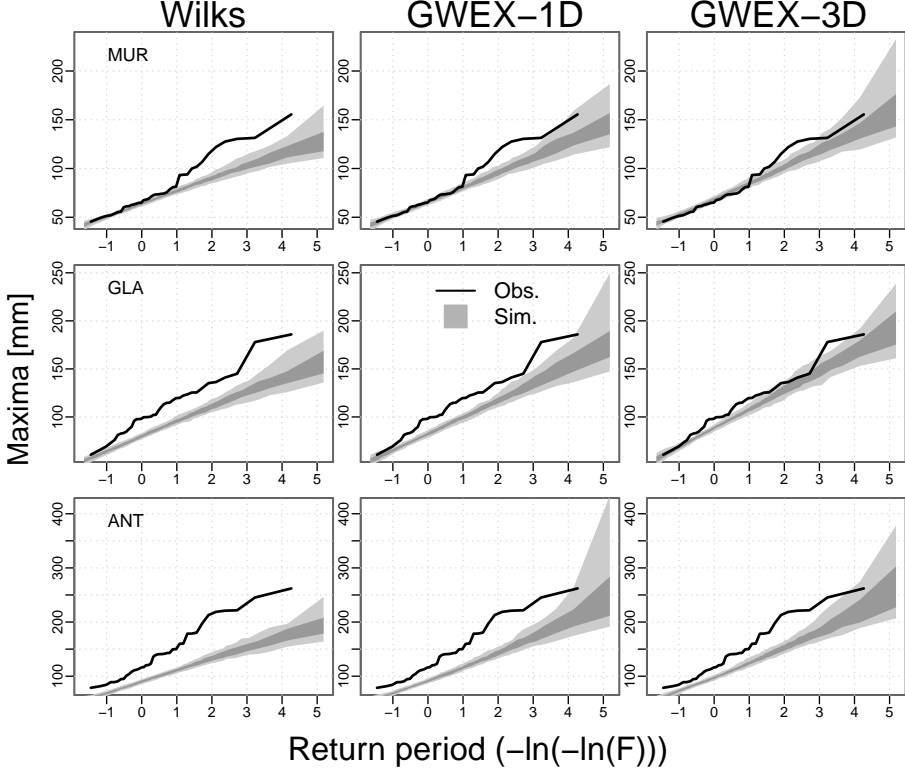

**Figure 11.** Simulated and observed 3-day annual maxima at the stations: 50% and 90% probability limits are shown.

Observed and simulated annual maximum precipitation at largest spatial scales are shown in Figures 12 and 13, at the daily and 3-day scales, respectively. At a daily scale, a slight under-estimation of the maxima by the Wilks and GWEX-1D models can be suspected, the simulated maxima being larger with GWEX-3D, especially at the scale of the entire Aare river catchment (bottom plots). At a 3-day scale, a dramatic underestimation of the maxima can be observed with Wilks and GWEX-1D. The slight underestimation observed at the scale of the stations, especially for the Wilks model, is far more severe at larger spatial scales. GWEX-3D does not suffer from such shortcomings, which can probably be explained by its direct representation of the spatial dependence at the 3-day scale.





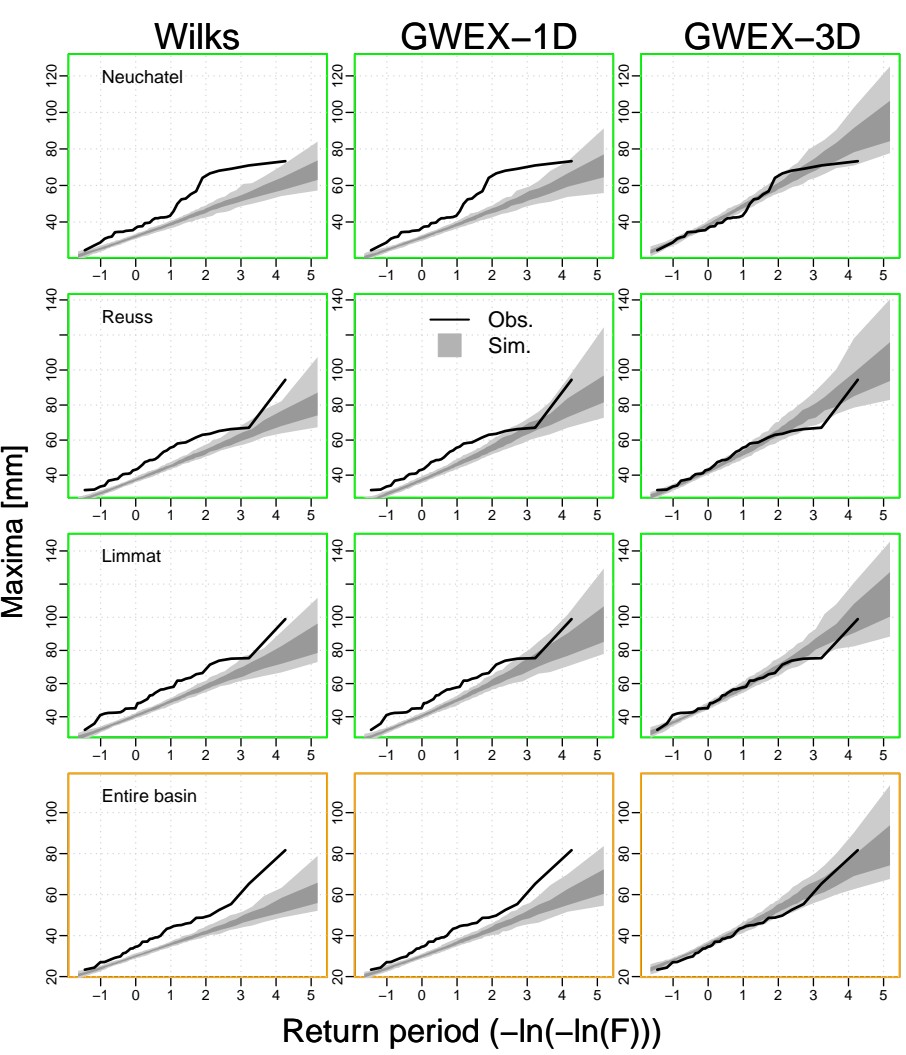

**Figure 12.** Simulated and observed daily annual maxima at the scale of the basins: 50% and 90% probability limits are shown.





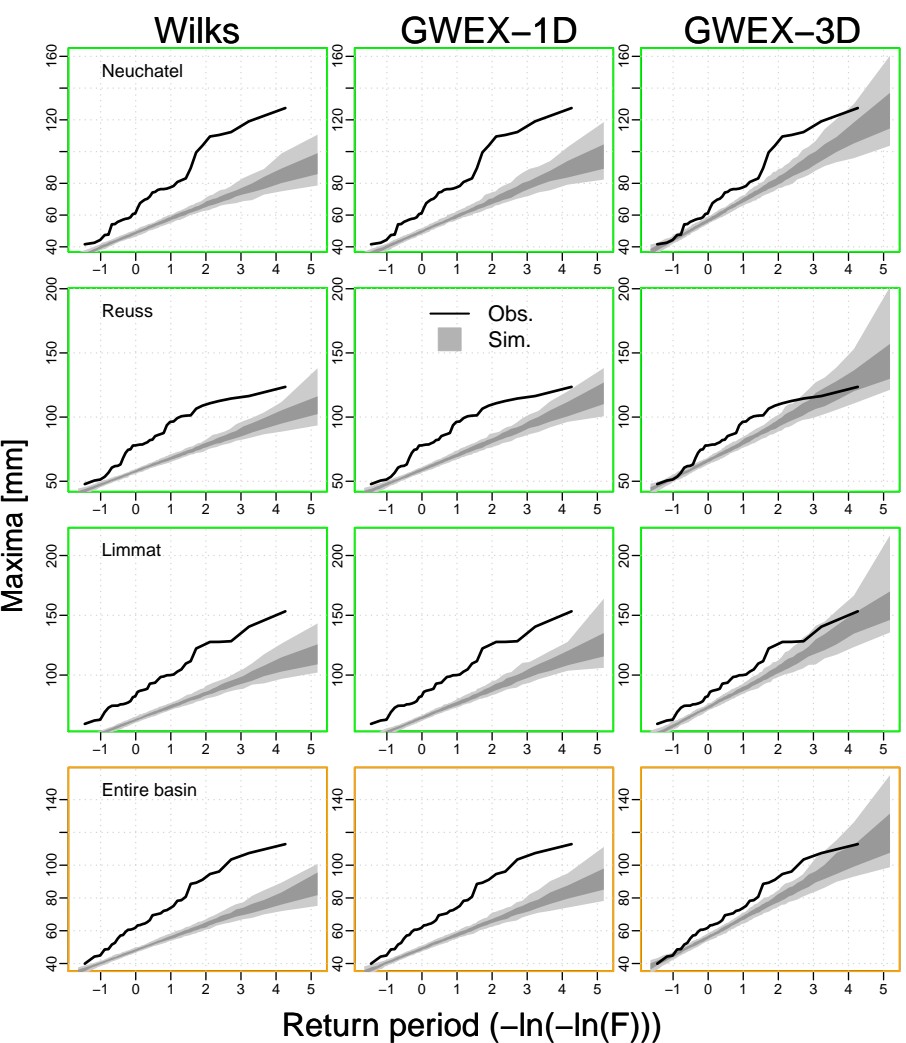

**Figure 13.** Simulated and observed 3-day annual maxima at the scale of the basins: 50% and 90% probability limits are shown.




Figures 14 and 15 show the observed and simulated 10-year and 50-year return periods, at a daily and a 3-day scales, respectively, for all spatial scales. These return periods are estimated empirically using the Gringorten formula (Gringorten, 1963). These figures summarize the previous illustrations and provide a more synthetic view of the model performances regarding extreme precipitation amounts. At a daily scale, there is no major difference of performances between the three

5    models. For the 50-year return periods, the number of 'poor' performance cases is below 20% for all models. However, at the 3-day scale, the under-estimation of the maxima by Wilks and GWEX-1D, as previously discussed, is clearly highlighted.

For GWEX-3D, the strategy consisting in simulating 3-day precipitation amounts, which are then disaggregated at a daily scale, presents several advantages:

– The model being fitted at a 3-day scale, 3-day maxima are adequately reproduced.

10    – As the method of fragments uses observed 3-day distributions to disaggregate 3-day amounts, the daily amounts resulting from a generated 3-day maxima are physically plausible. In particular, the temporal and spatial structures of large and persistent observed precipitation events are employed, which brings a coherence between the generated extreme events at the daily and 3-day scales.





**Figure 14.** Daily annual maxima for all spatial scales: 10-year (top plots) and 50-year (bottom plots) return periods. 90% probability limits are shown. The overall performance represents a percentage of all spatial scales.

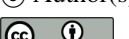





**Figure 15.** 3-day annual maxima for all spatial scales: 10-year (top plots) and 50-year (bottom plots) return periods. 90% probability limits are shown. The overall performance represents a percentage of all spatial scales.





# 5   Conclusions and outlook

The motivation for the development of precipitation models is usually the risk assessment of natural disasters (e.g. droughts, floods). The majority of existing precipitation models aims at reproducing a wide range of statistical properties of precipitation, at different scales, in order to be used as a general tool in different contexts. In this study, our main objective is to provide a

precipitation generator used in combination of an hydrological model for the evaluation of extreme flooding events, in a region covering approximately half of Switzerland. As a consequence, we are especially interested in the reproduction of extreme precipitation amounts at medium to large spatial scales. As the daily and 3-day precipitation amounts are a major determinant of the flood magnitude in large Swiss catchments (Froidevaux et al., 2015), an adequate reproduction of precipitation at these time scales is also required.

In this paper, we thus develop a multi-site precipitation model targeting the reproduction of extreme amounts at multiple temporal (daily, 3-day) and spatial scales. Two versions are considered, which are both based on the structure proposed by Wilks (1998). The first model version, GWEX-1D, enhance existing multi-site precipitation models using recent advances regarding extreme precipitations. In particular, an heavy-tailed distribution, the E-GPD, is applied to the precipitation intensities at each station. Temporal and spatial dependencies of the occurrence and intensity process are introduced using the copula theory and

a multivariate autoregressive process. In the second model version, GWEX-3D, the same structure is applied, but at a 3-day scale. 3-day simulated amounts are then disaggregated using an adaptation of the method of fragments (Buishand, 1991).

GWEX-1D and GWEX-3D are compared to the multi-site precipitation model proposed by Wilks (1998). The application of a multi-scale evaluation framework leads to the following conclusions:

- A 4-order Markov chain outperforms a first-order Markov chain for the transitions between dry and wet states, notably
in terms of reproduction of dry spell lengths.

- For winter months, the inter-annual variability of monthly aggregated amounts is clearly under-estimated by all the models. This under-estimation is not observed for summer months.

- At the scale of the stations, daily amounts (average, standard deviations and extremes) are reasonably well reproduced by all models.

- At a 3-day scale, precipitation extremes are severely under-estimated by Wilks and GWEX-1D. This under-estimation is observed at all spatial scales but is more pronounced at larger spatial scales.

As the GWEX-3D model outperforms the other precipitation models tested in this study, this is our recommended model for the evaluation of extreme flood events.

In this study, we support the arguments in favor of a systematic evaluation framework. The CASE framework proposed
by Bennett et al. (2017) provides useful tools in this respect, making possible a fair comparison of performances between precipitation models. Regarding the reproduction of extreme precipitation, we notice that evaluations are usually qualitative (e.g. one or two examples are provided and interpreted), and limited in terms of spatial scales (often at the stations only). The



evaluation of extreme precipitation amounts proposed in this paper is multi-scale in time (daily and 3-day scale) and in space (at the stations, for two different dissections of the study area, and for the entire Aare river catchment). Illustrative examples of the reproduction of annual maxima are supplemented with synthetic representations of these performances.

A possible enhancement of the GWEX-3D is the improvement of the inter-annual variability produced by the model. The solution proposed by Mehrotra et al. (2012), which consists in using a predictor based only on the aggregated number of precipitation occurrences over the previous 365 days, seems promising. Indeed, it avoids the introduction of atmospheric predictors, and preserve the purely stochastic behavior of the model. Future research will investigate if the floods resulting from generated precipitation scenarios through an hydrological model have statistical properties in agreement with observed floods.





## Appendix A:  Temporal disaggregation from a 3-day scale to a daily scale

For a 3-day period $\mathbf{D} = \{d, d+1, d+2\}$ starting on a day $d$, the observed and simulated precipitation amounts at a station $k$ are denoted by $Y_{\mathbf{D}}(k)$ and $\tilde{Y}_{\mathbf{D}}(k)$, respectively. We want to disaggregate the simulated 3-day amount for the period $\tilde{\mathbf{D}} = \{\tilde{d}, \tilde{d}+1, \tilde{d}+2\}$. This disaggregation is achieved with the application of the following steps:

1. A set of observed 3-day amounts are retained as candidate periods $\mathbf{D}$ according to two criteria:

   – **Season:** Periods $\tilde{\mathbf{D}}$ and $\mathbf{D}$ must belong to the same season, as defined in Section 3.2.

   – **Mean intensity:** Simulated and observed precipitation fields must have the same order of magnitude. Let $q_{0.5}$, $q_{0.75}$ and $q_{0.9}$ denote the quantiles of the mean observed precipitation intensities over all the stations associated to probabilities 0.5, 0.75 and 0.9, respectively. Observed and simulated 3-day periods are classified in 4 groups according to their mean intensity $\bar{Y} = \frac{1}{n}\sum_k Y_{\mathbf{D}}(k)$: dry periods ($\bar{Y} < q_{0.5}$), moderately wet periods ($q_{0.5} \leq \bar{Y} < q_{0.75}$), wet periods ($q_{0.75} \leq \bar{Y} < q_{0.9}$) and very wet periods ($q_{0.9} \geq \bar{Y}$).

   This first selection of candidate periods aims at increasing the chance of retaining periods corresponding to similar meteorological events.

2. For each observed 3-day candidate period $\mathbf{D}$, we compute the following score:

$$SCORE(\tilde{\mathbf{D}}, \mathbf{D}) = \sum_k \left| \frac{\tilde{Y}_{\tilde{d}-1}(k)}{\sum_k \tilde{Y}_{\tilde{d}-1}(k)} - \frac{Y_{d-1}(k)}{\sum_k Y_{d-1}(k)} \right| +$$
$$\left| \frac{\tilde{Y}_{\mathbf{D}}(k)}{\sum_k \tilde{Y}_{\mathbf{D}}(k)} - \frac{Y_{\mathbf{D}}(k)}{\sum_k Y_{\mathbf{D}}(k)} \right|.$$

This score measures the similarity between the simulated spatial field for the period $\tilde{Y}_{\mathbf{D}}(k)$ and the observed spatial field for the period $\tilde{\mathbf{D}}$, but also take into account the similarity between the spatial fields for the previous days $\tilde{d}-1$ and $d-1$.

Absolute differences between relative precipitation intensities are computed, which means that the lowest scores are obtained for spatial fields with similar shapes, among the observed periods corresponding to the same season and order of magnitude selected at the previous step.

3. For each simulated period $\tilde{\mathbf{D}}$, the observed precipitation fields corresponding to the 10 lowest scores are retained. For each station $k$, if a positive precipitation amount has been simulated ($\tilde{Y}_{\tilde{\mathbf{D}}}(k) > 0$), we look at the corresponding observed amount $Y_{\mathbf{D}}(k)$. If $Y_{\mathbf{D}}(k) = 0$, this observed period cannot be used to disaggregate $\tilde{Y}_{\tilde{\mathbf{D}}}(k)$ and we look at the next best observed field among the 10 selected fields. If the observed field contains a positive precipitation amount at this station ($Y_{\mathbf{D}}(k) > 0$), then we obtain the simulated daily amount for day $\tilde{d}$:

$$\tilde{Y}_{\tilde{d}}(k) = Y_d(k) \times \frac{\tilde{Y}_{\tilde{\mathbf{D}}}(k)}{Y_{\mathbf{D}}(k)}, \tag{A1}$$





with similar expressions for days $\tilde{d}+1$ and $\tilde{d}+2$. Simulated daily amounts correspond to the observed daily amounts, rescaled by the ratio between the simulated 3-day amount and observed 3-day amount. 3-day simulated amounts and observed temporal structures are thus preserved.

4. While the 3-day spatio-temporal coherence is generally conserved by applying the preceding steps, it can happen that the simulated 3-day amount is positive but there is no positive precipitation among the 10 best 3-day observed fields. In this case, we seek similar observed amounts at this station only, and randomly choose one 3-day period among the 10 best 3-day periods.

*Acknowledgements.* Financial support for this study by the Swiss Federal Office for Environment (FOEN), the Swiss Federal Nuclear Safety Inspectorate (ENSI), the Federal Office for Civil Protection (FOCP) and the Federal Office of Meteorology and Climatology, MeteoSwiss, through the project EXAR ("Evaluation of extreme Flooding Events within the Aare-Rhine hydrological system in Switzerland"), is gratefully acknowledged.





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
