# Peer review of "Stochastic generation of multi-site daily precipitation focusing on extreme events"

_Hydrology and Earth System Sciences, 2017_

## Referee Comment (RC1) · Anonymous Referee #1 · 6 Jun 2017

**1. Overview**

In the manuscript, the authors compare three different types of daily precipitation generators for the application in flood hazard modelling. One model is the so-called "Wilks-model", which was published in 1998 in the Journal of Hydrology (Wilks, 1998) and has served as the basis for a larger number of multi-site precipitation generators. The other two are new developments and called GWEX-1D and GWEX-3D. While GWEX-1D is simulating at daily scale, GWEX-3D simulates precipitation first at three days scale and then uses disaggregation based on observations to achieve daily precipitation as output. The three models are then compared with respect to different temporal and spatial

statistics.

2. General comments

- The title of the paper is a bit misleading. The three models may be used for the spatial assessment of floods and hydrological modelling is mentioned not only in the title, but also throughout the manuscript. However, the precipitation models are not applied in an impact assessment in this study and for this reason in my eyes the title should solely contain the comparison of three precipitation models. It is a bit irritating that the authors refer to the importance of several aspects of the precipitation model performance whose importance is not really demonstrated.

- The names of the new precipitation models are a bit misleading. First, "1D" and "3D" give the impression of any type of one- and three-dimensional simulation methodology. However, they represent days ("D"). I would rename the models into something more suitable.

- As far as I understand from the paper, the new GEWX models are actually "Wilks models" but with a new method to simulate the precipitation amounts (using temporally and spatially correlated random numbers from an autoregressive process and using a Student copula for the spatial component). I think this should be stated as such in the paper as the manuscript presents the new models more as a revolution rather than an evolution. So one of the first sentences could be that the paper deals with two modifications of the Wilks approach.

- The motivation behind the study and for the new model developments is the impact assessment. However, without the same the reader will not be able to really understand the sensitivity of certain statistics in regard to the assessment of extreme floods. I think the importance of some of the statistical metrics should be explained in more detail referring to the area of their application, and proof must be given of their relevance. Other literature in such a study (complex mountain region) is not very convincing to me.

[Figure]

- The abstract is incomplete and must be much more detailed and specific. What is an "event"? What is "large"? What are "recent advances"? The abstract should mention the Wilks model, the two new models (maybe also a short sentence how they work) and the basic outcomes of the study.

- The Wilks model could likewise be applied with E-GPD distributions for precipitation intensities and Markov chains of the order 4. That is, revealed weaknesses of the Wilks model can easily be addressed. I recommend adapting the Wilks approach for a more objective comparison. The original Wilks approach is not a given and was just one application for a specific dataset in the US and in my eyes it should always be revised for other study areas and climates.

- For flood modelling, the lagged cross correlations (see Wilks 1998, page 183) can be very important as they represent the progression of weather systems across the study area. Especially at larger scales the progression of weather events may be important. I strongly recommend plotting these statistics for all three models.

- The autocorrelation of precipitation is addressed by MAR(1) models in the GWEX models. I would recommend plots for the autocorrelation of the precipitation intensities for some sites to see potential differences in their performance.

3. Specific comments

- Line 8. I think there is a language issue.

- Line 10. Not only conceptual models. There are more recent studies for coupling WGs with impact models.

- Page 1 bottom/ Page 2 top: In my eyes the classification is not fully correct. All these models are multi-site models. Also resampling methods are multi-site models. I recommend a more suitable classification even though I admit that the variety of the existing multi-site models makes a clear classification more and more difficult (also the authors combined parametric and non-parametric techniques).

- Page 4, Line 8. Are Thiessen polygons suitable for such a complex mountainous study region?

- Page 7 "Marginal distributions". Can any proof be given that the more complex fitting of a combined distribution is really significantly better for the simulation of the extremes in this region? Also here, the most prominent argument is other literature.

- Page 9, top of the page. If the Gaussian copula is not suitable for simulating spatially dependent extremes but the Student copula is, this could be demonstrated. I am thinking of readers who want to build the code but are not experts in copulas and want to understand the significance.

- Page 9 bottom. Why are Markov chains of the order 4 used? Have there been statistical tests or sensitivity studies to underline this decision? Later on, some remarks are given on the simulation of short dry spells, but I think this should be addressed in a more structured way.

- Page 11, Table 11 (and figures). Red and green are not suitable for figures, please change the colours as some people cannot read them otherwise (https://www.nature.com/nature/journal/v510/n7505/full/510340e.html).

- Page 12, Line 28. I guess it is very difficult to say if an extreme precipitation amount is unrealistic or not as long as they are physically possible?

- Page 16, Line 18-20. If the order of the Markov chain is the issue for short dry spells, this can be easily adapted by using the same order in the original Wilks approach. What was the argument for using the first order Markov chains in the Wilks model? (see comment above)

- Page 21, Line 8-9. Please explain the seasonal differences with explicit reference to the study area and its climatology for better understanding.

- Page 22, section 4.4. and figure 10. To me, the performance looks fair for all three models. The main difference is the simulation of higher extremes with the GEWX

models. The authors mention the difference but it needs further discussion. Also, how can we know that the extremes of one method are more realistic than from another? While we know little about the validity of the simulated extremes, they may have a big impact on simulated floods, especially in small catchments (but as mentioned before, this is not examined in the paper).

- Page 26 Line 10-13. It is not surprising that the non-parametric disaggregation leads to a better performance. I understand its strengths but it may likewise be a limiting factor in generating extremes.

- Page 20, first line 2-9. As already mentioned, I see the motivation behind the study (and it is generally a good one). But without any proof that the differences in the performance of the three precipitation models really have a significant impact on the simulation results of hydrological extremes (also considering all the uncertainties in hydrological models), the significance of the research outcomes remain questionable.

- Page 29, Line 21-22. Please explain why, see comments above.

- Page 29, Line 27-28. The issue of larger spatial scales could be addressed by running more analyses at smaller scales. So the key motivation of the study is probably to examine large flood events and their spatial dependences? If so, this should be better explained. But again, without really simulating the floods throughout different scales the arguments for a particular precipitation model choice is questionable.

- Page 30. Is the underestimation of the inter-annual variability such a big issue in Switzerland and for flood modelling? I would assume it is more an issue in more arid regions and for example agricultural studies? Some more remarks on the relevance in Switzerland and floods in general would be useful.

4. Summary of review

- The abstract needs revision and must be more detailed (see general comments).

- The introduction is not very well structured. The arguments for the construction of

the new precipitation methodologies are mainly based on other literature and reasoning. The context of the paper should (i) either be revised (comparison of precipitation models) or (ii) proof must be given of the advantages using the new models by really coupling them with a hydrological model and examining the estimated flood events in the study region. I think it is the key weak point of the paper: reference is given to an application, which is not really done. Also, the title and abstract are a bit misleading and the reader may expect a flood modelling study and thus more than what has been presented.

- For the three different precipitation models, I would recommend a flow chart with the Wilks model as the central component and then the adaptations that have been done. This makes it easier for the reader to understand all models and what has been changed.

- Although the level of English is very good, some (minor) mistakes can be found in the manuscript and a native speaker should probably have a final look before resubmission.

5. Literature

Wilks, D. S.: Multisite generalization of a daily stochastic precipitation generation model, Journal of Hydrology, 210, 178–191, 1998.

---

## Referee Comment (RC2) · Anonymous Referee #2 · 9 Jun 2017

The authors propose extensions of a classical multisite daily rainfall generator initially proposed by Wilks in 1998. The framework of Wilks model is flexible enough to allow many adaptations, and the authors of this paper propose - to add more structure in the dynamics of the model by considering higher order Markov model for the occurrence process and an autoregressive component for the amounts - to use a hybrid distribution for the marginal distribution to deal with heavy tail distributions - to use a Student copula for the spatial structure to catch upper tail dependance I believe that all these extensions make sense and are interesting to try.

General comments

Many extensions of the Wilks model have already been proposed in the literature. I think that a review of this literature must be included in the paper and that the authors should explain why the extension that they propose is original and useful with respect to this literature.

In my opinion, one weakness of the paper is that the model is formulated as a simulation tool rather than as a proper statistical model. It is also the case for the original Wilks model, but it has then been reformulated by other authors as a statistical model, see e.g. Thompson et al. (2007). I think that the paper would be easier to read for statisticians like me if a similar formalization was done in the paper. In particular, the various assumptions on the occurrence/amount processes should be written precisely using formulas and the definition of the model should be separated from the discussion on parameter estimation and simulation.

I believe that the validation part must also be improved. First, some usual validation criteria for rainfall generators, such as diagnostics based on the marginal distribution (e.g. qqplot) and the second order structure of the process (autocorrelation and cross-correlation functions) are not shown and it makes it difficult to see the benefit of using a hybrid distribution and the autoregressive component. Also the chosen validation criteria does not permit to see the interest of using a student Copula (does it really improve the modeling of extremal dependance?). Finally, I find the simulation results generally disappointing. If I understand correctly the categorization, we should obtain about 90% of good if the model was able to reproduce the statistics of the observed rainfall? Is it satisfactory to obtain percentage around 50%?

Specific comments

- Keywords are missing?

- End of Page 1/top of page 2. I am not really satisfied by the proposed classification. For example weather type models are often used as multisite rainfall generators (without conditioning to large scale information). Also it would be useful to cite the review

papers on rainfall generators here.

- Section 2.1. The authors go directly from a Markov chain of order p=1 to a Markov chain of order p=4. I would expect that the best value of p is somewhere between these two values. The authors could try to find the optimal value of p, using for example standard model selection criteria.

- Equation (5). I am surprised that the authors use a diagonal matrix for A. I would expect that it is useful to add some spatial structure here?

- Section 2.3 and 3.3 should be merged.

- Section 3. Why is it called "Application"? I do not see any application here.

References

Thompson, C. S., Thomson, P. J., & Zheng, X. (2007). Fitting a multisite daily rainfall model to New Zealand data. Journal of Hydrology, 340(1), 25-39.

---

## Author Comment (AC1) · 14 Jun 2017

**Response to Interactive comment by Anonymous Referee #1**

*We thank the referee for this thorough review and for the numerous constructive suggestions. We agree that the general presentation can be improved and these suggestions will be incorporated in the modified manuscript.*

[Figure]

**1   General comments**

1.1. The title of the paper is a bit misleading. The three models may be used for the spatial assessment of floods and hydrological modelling is mentioned not only in the title, but also throughout the manuscript. However, the precipitation models are not applied in an impact assessment in this study and for this reason in my eyes the title should solely contain the comparison of three precipitation models. It is a bit irritating that the authors refer to the importance of several aspects of the precipitation model performance whose importance is not really demonstrated.

*We agree that the title can be misleading and could be replaced by 'Stochastic generation of multi-site daily precipitation focusing on extreme events'. We think that it is important to indicate the emphasis on the reproduction of very large precipitation events, in terms of intensity, duration, and spatial extent.*

1.2.   The names of the new precipitation models are a bit misleading.   First, "1D" and "3D" give the impression of any type of one- and three-dimensional simulation methodology. However, they represent days ("D"). I would rename the models into something more suitable.

*This is a good suggestion and the names will be replaced by:*

1. *Wilks: the current 'Wilks' model,*

2. *Wilks_EGPD: A modified Wilks version, with the EGPD and a Markov chain of order 4, as suggested by the referee (see comment #1.6),*

3. *GWEX: the current GWEX-1D,*

4. *GWEX_Disag: the current GWEX-3D. It would indicate more clearly the disaggregation step which follows up the simulation at a 3-day scale.*

1.3. As far as I understand from the paper, the new GWEX models are actually "Wilks models" but with a new method to simulate the precipitation amounts (using temporally and spatially correlated random numbers from an autoregressive process and using a Student copula for the spatial component). I think this should be stated as such in the paper as the manuscript presents the new models more as a revolution rather than an evolution. So one of the first sentences could be that the paper deals with two modifications of the Wilks approach.

*We agree. The fact that GWEX are evolutions of the Wilks model must be clearly stated. In fact, it is already indicated at p.2/l.25 and p.4/l.10 and throughout the presentation of the models. As suggested by the referee, we will also indicate this point directly in the abstract. However, it must also be underlined that GWEX is a significant evolution of the Wilks model. First, as underlined by the referee, the methodology applied to simulate the precipitation amounts is considerably modified. We consider different temporal and spatial dependences, and we also discuss the choice of the marginal distribution in details, which is currently overlooked in the literature of precipitation stochastic models. Second, GWEX-3D (which will be named GWEX_Disag) combines simulations at a 3-day scale and a disaggregation approach, which represents a further step in the complexity of the model. In our opinion, GWEX cannot only be considered as a slight modification/evolution of the Wilks model.*

1.4. The motivation behind the study and for the new model developments is the impact assessment. However, without the same the reader will not be able to really understand the sensitivity of certain statistics in regard to the assessment of extreme floods. I think the importance of some of the statistical metrics should be explained in more detail referring to the area of their application, and proof must be given of their relevance. Other literature in such a study (complex mountain region) is not very convincing to me.

*We thank the referee fo this suggestion and additional details regarding the importance of the statistical metrics will be provided in the modified version. In particular,*

[Figure]

*Froidevaux (2014) analyze meteorological events triggering floods in Switzerland. These studies have been very briefly mentioned at the beginning of p.12 and these results must be discussed in more details, as will be done in the revised manuscript. However providing a proof of the relevance seems complicated without the hydrological application (which is clearly beyond the scope of this paper, as discussed in comment #2.14.). If the referee has more specific metrics that could be presented, we would be glad to include them in our study.*

1.5. The abstract is incomplete and must be much more detailed and specific. What is an "event"? What is "large"? What are "recent advances"? The abstract should mention the Wilks model, the two new models (maybe also a short sentence how they work) and the basic outcomes of the study.
*We thank the referee for this constructive suggestion. Additional details will be added to the abstract.*

1.6. The Wilks model could likewise be applied with E-GPD distributions for precipitation intensities and Markov chains of the order 4. That is, revealed weaknesses of the Wilks model can easily be addressed. I recommend adapting the Wilks approach for a more objective comparison. The original Wilks approach is not a given and was just one application for a specific dataset in the US and in my eyes it should always be revised for other study areas and climates.
*We thank the referee for this suggestion. An additional version of the Wilks model, with the EGPD instead of a mixture of exponential distributions for the marginal distributions, and a Markov chain of order 4 instead of order 1, will be presented in the modified manuscript.*

1.7. For flood modelling, the lagged cross correlations (see Wilks 1998, page 183) can be very important as they represent the progression of weather systems across the study area. Especially at larger scales the progression of weather events

may be important. I strongly recommend plotting these statistics for all three models.
*We appreciate this judicious suggestion and lagged cross correlation will be added to the set of statistics presented in the paper.*

1.8. The autocorrelation of precipitation is addressed by MAR(1) models in the GWEX models. I would recommend plots for the autocorrelation of the precipitation intensities for some sites to see potential differences in their performance.
*We thank the referee for this suggestion. Plots for the autocorrelation of the precipitation amounts will be shown for some of the stations.*

**2 Specific comments**

2.1. Line 8. I think there is a language issue.
*This sentence will be reformulated in the revised version.*

2.2. Line 10. Not only conceptual models. There are more recent studies for coupling WGs with impact models.
*Thanks for this remark. References about these recent studies can be included in the revised version.*

2.3. Page 1 bottom/ Page 2 top: In my eyes the classification is not fully correct. All these models are multi-site models. Also resampling methods are multi-site models. I recommend a more suitable classification even though I admit that the variety of the existing multi-site models makes a clear classification more and more difficult (also the authors combined parametric and non-parametric techniques).
*Here, multi-site models refer to models that target the reproduction of statistics at*

*specific sites. They can be opposed to random fields that mainly intend to reproduce spatial properties (e.g. the variogram). We agree that resampling methods are also referred as multi-site models in the literature (see, e.g. Mehrotra et al., 2006). Hence, we propose to replace 'multi-site models' at line 12 by 'statistical multi-site models' in order to clarify the distinction between resampling methods and the set of models cited in the paragraph 'multi-site models' which applies various statistical structure (copulas, truncated Gaussian distributions, V-copula transform, etc.).*

2.4. Page 4, Line 8. Are Thiessen polygons suitable for such a complex mountainous study region?
*The computation of areal precipitation values is a difficult task considering the spatial and temporal variability of precipitation events, the complex topography of the sudy area, and the limited number of pluviographs. In Switzerland, Schäppi (2013) shows that the topography impacts rainfall amounts differently according to the type of meteorological event. In a preliminary study, the impact of different interpolation methods (inverse distance, ordinary kriging, kriging with external drift, Thiessen polygons) and different sets of stations (399, 211, 129, 47 and 22 stations) on extreme areal precipitation amounts has been analyzed. The main conclusion was that the number of stations was a much more important factor than the interpolation method. This was the main motivation for the application of the stochastic models to a high number (105) of stations. Furthermore, it is important to notice that applying more complex interpolation methods (e.g. kriging methods) increase significantly the computational cost, which can be prohibitive for the production of long meteorological scenarios.*

2.5. Page 7 "Marginal distributions". Can any proof be given that the more complex fitting of a combined distribution is really significantly better for the simulation of the extremes in this region? Also here, the most prominent argument is other literature.
*QQ-plots will be provided in the revised version in order to assess the quality of the*

*fitting of these marginal distributions. However, it is very important to note that local applications give limited proof regarding the performance of a distribution for the fitting of extreme values. As indicated in "Papalexiou, S. M. and Koutsoyiannis, D. (2013) Battle of extreme value distributions: A global survey on extreme daily rainfall, Water Resources Research, 49, 187201", most studies of extreme rainfall are inconclusive because they are too specific to particular areas or stations. The main explanation for these failures is that fitting and inferring the distribution tails is subject to high uncertainties in the estimation of the parameters, even for long time series (this point is also discussed and illustrated in Evin et al., 2016). The references given in the paper (Papalexiou and Koutsoyiannis, 2013; Serinaldi and Kilsby, 2014) are conclusive precisely because they are the result of a very large number of applications, and give strong arguments in favor of the application of heavy-tailed distributions. Furthermore, Figures 2 and 3 prove that low tail-distributions (like a mixture of exponentials) would lead to an under-estimation of extreme precipitations in some regions (regions where $\xi$ is different from 0, in green, yellow and red).*

2.6. Page 9, top of the page. If the Gaussian copula is not suitable for simulating spatially dependent extremes but the Student copula is, this could be demonstrated. I am thinking of readers who want to build the code but are not experts in copulas and want to understand the significance.
*With an additional version of the Wilks model, with the EGPD instead of a mixture of exponential distributions for the marginal distributions (see comment #1.6.), we will be able to assess the difference between a Gaussian copula and a Student copula for the reproduction of daily precipitation extremes.*

2.7. Page 9 bottom. Why are Markov chains of the order 4 used? Have there been statistical tests or sensitivity studies to underline this decision? Later on, some remarks are given on the simulation of short dry spells, but I think this should be addressed in a more structured way.

*At p.5/l.10, we indicate that Srikanthan and Pegram (2009) apply a 4-order Markov chain and show that it improves the reproduction of dry/wet period lengths. This point will be reminded at p.9.*

2.8. Page 11, Table 11 (and figures). Red and green are not suitable for figures, please change the colours as some people cannot read them otherwise (https://www.nature.com/nature/journal/v510/n7505/full/510340e.html).
*We thank the referee for this comment. These colors will be modified to be more suitable for most color-blind people, (https:// www.nature.com/ nmeth/ journal/ v8/ n6/ full/ nmeth.1618.html). As we understand this issue, it seems that types of green (bluish green) and red (vermillion) are more adapted to color-blind individuals.*

2.9. Page 12, Line 28. I guess it is very difficult to say if an extreme precipitation amount is unrealistic or not as long as they are physically possible?
*It is true that an extreme precipitation amount cannot be considered as unrealistic if the amount is physically possible. However, it is difficult to define what amount can be considered as impossible. Here, we indicate that large $\xi$ parameters ($> 0.25$) lead to extremely heavy-tail distributions. In practice, very large daily amounts (e.g. $> 1000$ mm) will often be obtained when long runs are performed (e.g. 1000 years). The reference given in the paper (Serinaldi and Kilsby, 2014) indicates that these large $\xi$ parameters are often spurious as they usually are the results of high parameter uncertainties. Furthermore, for $\xi > 1/3$, the Generalized Pareto distribution has an infinite variance, which is not a desirable properties. Note that this constraint has a limited impact in our case since we always obtain $\xi < 0.25$ in our study area (see dark red areas in Fig. 2 and 3). However, we think that it is important to inform potential users of GWEX (and more generally anyone who applies a GPD to extreme precipitation amounts) that they need to be very careful if they obtain very high $\xi$ estimates.*

2.10. Page 16, Line 18-20. If the order of the Markov chain is the issue for

short dry spells, this can be easily adapted by using the same order in the original Wilks approach. What was the argument for using the first order Markov chains in the Wilks model? (see comment above)
*See comment #2.7.*

2.11. Page 21, Line 8-9. Please explain the seasonal differences with explicit reference to the study area and its climatology for better understanding.
*More explanations about these seasonal differences, specific to Switzerland, wil be provided in the revised version.*

2.12. Page 22, section 4.4. and figure 10. To me, the performance looks fair for all three models. The main difference is the simulation of higher extremes with the GEWX models. The authors mention the difference but it needs further discussion. Also, how can we know that the extremes of one method are more realistic than from another? While we know little about the validity of the simulated extremes, they may have a big impact on simulated floods, especially in small catchments (but as mentioned before, this is not examined in the paper).
*We agree with the referee, the performance looks fair for all three models. However, this figure only points out differences of behavior between the three models. As mentioned in the comment #2.5., these illustrative examples cannot be used to test the performances of the different models for the simulations of extreme daily precipitation amounts. The only way to perform such a validation is to apply metrics on a large set of applications (here, for example, at all the stations), which is done at Figures 14 and 15. This remark will be added to the revised version of the manuscript.*

2.13. Page 26 Line 10-13. It is not surprising that the non-parametric disaggregation leads to a better performance. I understand its strengths but it may likewise be a limiting factor in generating extremes.
*In our opinion, GWEX-3D (which will be named GWEX_Disag) represents the best*
*combination between a purely statistical approach and a nonparametric approach. The results presented in this study confirm this statement. As we know little about very extreme precipitations, it is, in our opinion, impossible to know if it is a limiting factor or not.*

2.14. Page 29, first line 2-9. As already mentioned, I see the motivation behind the study (and it is generally a good one). But without any proof that the differences in the performance of the three precipitation models really have a significant impact on the simulation results of hydrological extremes (also considering all the uncertainties in hydrological models), the significance of the research outcomes remain questionable. *The rational behind the study is the following: we want to develop a stochastic model for precipitation which preserve the most critical properties of precipitation at different spatial and temporal scales. Froidevaux (2014) shows that extreme 3-day precipitation amounts often trigger important floods. In this context, it seems natural to target the reproduction of 3-day precipitation amounts. However, we agree that hydrological applications would validate the importance of such properties. Actually, hydrological applications are currently undertaken by the University of Zürich. A conceptual hydrological model (HBV) is applied to 87 sub-basins partitioning the whole study area, using precipitation scenarios produced by GWEX as inputs. Numerous technical issues still need to be resolved. Some basins are ungauged, or with very short stream-flow series. The hydrological system of the Aare-Rhine river needs to be treated as a whole since floods at larger spatial scales need also to be investigated. Rating curves have very high uncertainties in some basins and need to be re-evaluated. All these aspects have to be treated in details. It is important to note that the hydrological study (as well as our study) is particularly challenging considering the large spatial extent of the Aare river catchment. These studies stand out from similar studies which are usually limited to e few precipitation stations and one "small" catchment (see, e.g., Keller et al., 2015, recently published in HESS, with an application to 8 precipitation stations located in a catchment with a size of 1700 km$^2$, to be compared with our*

*study area of 17,000 km$^2$).* Clearly, the hydrological application should be presented in future publications, considering the complexity of this work and the amount of results. However, we agree that the hydrological application would emphasize the significance of this study, and this point must be discussed in the manuscript.

2.15. Page 29, Line 21-22. Please explain why, see comments above.
*See Comment # 3.11.*

2.16. Page 29, Line 27-28. The issue of larger spatial scales could be addressed by running more analyses at smaller scales. So the key motivation of the study is probably to examine large flood events and their spatial dependences? If so, this should be better explained. But again, without really simulating the floods throughout different scales the arguments for a particular precipitation model choice is questionable.
*The key motivation is to develop a stochastic model for precipitation which preserve the most critical properties of precipitation at different spatial and temporal scales, and especially for extreme precipitation amounts. This motivation is rather general and not specific to some characteristics of flood events (e.g. their spatial dependence).*

2.17. Page 30. Is the underestimation of the inter-annual variability such a big issue in Switzerland and for flood modelling? I would assume it is more an issue in more arid regions and for example agricultural studies? Some more remarks on the relevance in Switzerland and floods in general would be useful.
*Thanks for this remark. We agree that more comments about the relevance of these metrics should be provided, which need to be more specific to applications in Switzerland*

**3   Summary of review**

3.1. The abstract needs revision and must be more detailed (see general comments). *See comment # 1.5.*

3.2.   The introduction is not very well structured.   The arguments for the construction of the new precipitation methodologies are mainly based on other literature and reasoning.   The context of the paper should (i) either be revised (comparison of precipitation models) or (ii) proof must be given of the advantages using the new models by really coupling them with a hydrological model and examining the estimated flood events in the study region. I think it is the key weak point of the paper: reference is given to an application, which is not really done.  Also, the title and abstract are a bit misleading and the reader may expect a flood modelling study and thus more than what has been presented.
*We agree that the introduction can be misleading and it will be modified in order to clearly indicate that this study mainly aims at comparing precipitation models, the hydrological context being the motivation for the thorough assessment of areal precipitation extremes.*

3.2. For the three different precipitation models, I would recommend a flow chart with the Wilks model as the central component and then the adaptations that have been done. This makes it easier for the reader to understand all models and what has been changed.
*This is an excellent suggestion and a flow chart will be added to the revised version of the manuscript.*

3.3.   Although the level of English is very good, some (minor) mistakes can be found in the manuscript and a native speaker should probably have a final look before resubmission.

*The revised version will be proofread by a native speaker.*

**References**

Evin, G., Blanchet, J., Paquet, E., Garavaglia, F., and Penot, D. (2016). A regional model for extreme rainfall based on weather patterns subsampling. *Journal of Hydrology*, 541, Part B:1185–1198.

Froidevaux, P. (2014). *Meteorological characterisation of floods in Switzerland*. PhD thesis, Geographisches Institut, University of Bern.

Keller, D. E., Fischer, A. M., Frei, C., Liniger, M. A., Appenzeller, C., and Knutti, R. (2015). Implementation and validation of a Wilks-type multi-site daily precipitation generator over a typical Alpine river catchment. *Hydrology and Earth System Sciences*, 19(5):2163–2177.

Mehrotra, R., Srikanthan, R., and Sharma, A. (2006). A comparison of three stochastic multi-site precipitation occurrence generators. *Journal of Hydrology*, 331(1–2):280–292.

Papalexiou, S. M. and Koutsoyiannis, D. (2013). Battle of extreme value distributions: A global survey on extreme daily rainfall. *Water Resources Research*, 49(1):187–201.

Schäppi, B. (2013). *Measurement and analysis of rainfall gradients along a hillslope transect in the Swiss Alps*. PhD thesis, ETH Zürich, Zürich, Switzerland.

Serinaldi, F. and Kilsby, C. G. (2014). Rainfall extremes: Toward reconciliation after the battle of distributions. *Water Resources Research*, 50(1):336–352.

Srikanthan, R. and Pegram, G. G. S. (2009). A nested multisite daily rainfall stochastic generation model. *Journal of Hydrology*, 371(1–4):142–153.

---

## Author Comment (AC2) · 14 Jun 2017

**Response to Interactive comment by Anonymous Referee #2**

The authors propose extensions of a classical multisite daily rainfall generator initially proposed by Wilks in 1998. The framework of Wilks model is flexible enough to allow many adaptations, and the authors of this paper propose - to add more structure in the dynamics of the model by considering higher order Markov model for the occurrence process and an autoregressive component for the amounts - to use a hybrid distribution for the marginal distribution to deal with heavy tail distributions - to use a Student

copula for the spatial structure to catch upper tail dependence. I believe that all these extensions make sense and are interesting to try.

*We thank the referee for this review and for these comments. Most of these suggestions will be incorporated in the modified manuscript.*

**1   General comments**

1.1. Many extensions of the Wilks model have already been proposed in the literature. I think that a review of this literature must be included in the paper and that the authors should explain why the extension that they propose is original and useful with respect to this literature.
*We agree that the differences between GWEX and the existing extensions of the Wilks models should be presented in the introduction. This discussion will be included in the revised version of the manuscript.*

1.2. In my opinion, one weakness of the paper is that the model is formulated as a simulation tool rather than as a proper statistical model. It is also the case for the original Wilks model, but it has then been reformulated by other authors as a statistical model, see e.g. Thompson et al. (2007). I think that the paper would be easier to read for statisticians like me if a similar formalization was done in the paper. In particular, the various assumptions on the occurrence/amount processes should be written precisely using formulas and the definition of the model should be separated from the discussion on parameter estimation and simulation.
*We thank the reviewer for this excellent suggestion. A more formal mathematical formulation of GWEX could certainly improve the presentation. We also agree that a specific section should be devoted to parameter estimation and model simulation. This will be done in the revised manuscript.*

[Figure]

1.3. I believe that the validation part must also be improved. First, some usual validation criteria for rainfall generators, such as diagnostics based on the marginal distribution (e.g. qqplot) and the second order structure of the process (autocorrelation and crosscorrelation functions) are not shown and it makes it difficult to see the benefit of using a hybrid distribution and the autoregressive component. Also the chosen validation criteria does not permit to see the interest of using a student Copula (does it really improve the modeling of extremal dependence?).

*These remarks have also been made by the referee #1 (comments #1.8, #2.5 and #2.6). QQ-plots will be provided to assess (visually) the quality of the fitting for the marginal distributions. Additional figures will also be provided to assess the performances concerning the autocorrelations at some stations and the reproduction of cross-correlations. Finally, an additional model version, a modified Wilks version with the EGPD, will be added to the current models, which will enable the assessment of the impact of the Student copula (versus a Gaussian spatial structure).*

1.4. Finally, I find the simulation results generally disappointing. If I understand correctly the categorization, we should obtain about 90% of good if the model was able to reproduce the statistics of the observed rainfall? Is it satisfactory to obtain percentage around 50%?

*Yes, we should obtain about 90% of good if the model is able to reproduce the observed statistics, and very few 'poor' cases. As indicated in the paper, our primary criteria to judge the overall performance of a model is the number of metrics for which 'poor' performances are obtained. We agree that these percentages are subjective (why 90%? Is 50% of good cases good enough?) but not more subjective, in our opinion, that the visual inspection of a QQ-plot. Furthermore, the purpose of the CASE framework, as presented in Bennett et al. (2017), is to enable a more systematic comparison of stochastic models. Our study also tries to promote this approach. A more systematic comparison of the models, which includes a consistent way to*

[Figure]

*compute the performance metrics, is important in order to obtain a fair assessment of the strengths/weaknesses of the different models. For this reason, this study applies the classification proposed by Bennett et al. (2017), without modifying the classification.*

**2 Specific comments**

2.1. Keywords are missing?
*In HESS, to the extent of our knowledge, keywords do not appear in the manuscript.*

2.2. End of Page 1/top of page 2. I am not really satisfied by the proposed classification. For example weather type models are often used as multisite rainfall generators (without conditioning to large scale information). Also it would be useful to cite the review papers on rainfall generators here.
*We agree that the terminology 'Multi-site models' is too vague here. A similar comment has been done by the referee #1 (see its comment #2.3.). We propose to replace 'multi-site models' at line 12 by 'statistical multi-site models'. Weather type models that are not conditioned by large scale information (for example using 'dry' and 'wet' states that are inferred) could belong to this class of models. We also agree that review papers could be included in the introduction and this will be done in the revised manuscript.*

2.3. Section 2.1. The authors go directly from a Markov chain of order p=1 to a Markov chain of order p=4. I would expect that the best value of $p$ is somewhere between these two values. The authors could try to find the optimal value of $p$, using for example standard model selection criteria.

*We thank the reviewer for this suggestion. It is true that an optimal value might be found if there was an easy selection criteria. As this point is not central in our study, a direct comparison of Markov chains of order $p = 1$ and $p = 4$ is deemed sufficient.*

2.4. Equation (5). I am surprised that the authors use a diagonal matrix for $A$. I would expect that it is useful to add some spatial structure here?
*Initial version of GWEX were applying a full covariance matrix for $A$. However, it seems that large covariance matrices are often very close to a non positive definite. This is not really problematic during the estimation step, but leads to very unstable results during the simulation step. As applying a diagonal matrix for $A$ does not degrade the performances of GWEX, this solution was retained.*

2.5. Section 2.3 and 3.3 should be merged.
*We thank the reviewer for this suggestion. Following comment #1.2., these sections will be re-organized with specific sections devoted to the estimation and the simulation steps. Sections 2.3 and 3.3 will thus be merged in the revised manuscript.*

2.6. Section 3. Why is it called "Application"? I do not see any application here.
*Following previous comments (comments #1.2. and 2.5.), the different parts of section 3 will be moved to other sections. Section 3.1 'Split-sampling procedure' is related to the evaluation framework. Section 3.2 'Regionalization of the $\xi$ parameter' is related to the estimation of the parameters. Section 3.3 'Generation of scenarios' will be merged to section 2.3 (see previous comment).*

**References**

Bennett, B., Thyer, M., Leonard, M., Lambert, M., and Bates, B. (2017). A comprehensive and systematic evaluation framework for a parsimonious daily rainfall field model. *Journal of Hydrology*. https://doi.org/10.1016/j.jhydrol.2016.12.043, In Press.

---

## Author Comment (AC3) · 27 Jun 2017

We thank the two anonymous referees for their constructive suggestions, comments and questions. We truly believe that they will lead to a significant improvement of the manuscript. This response provides a summary of these comments and of our answers.

[Figure]

**Overall presentation of the manuscript**

Most of the referee's comment are related to the presentation of the methodology and the results. These comments are entirely justified and are appreciated, as they will greatly enhance the paper. The following paragraphs summarize what modifications will be made to the manuscript (more details can be found in the interactive comments):

- **Abstract:** We agree with the referee #1 (comments #1.5. and #3.1.) that the current abstract is not specific enough. Additional details will be provided (summary of the model developments, key results, etc.)

- **Title and introduction misleading:** As pointed out by the referee #1 (comments #1.1. and #3.2.), the title and the introduction seem to indicate that our study shows the results of an hydrological application, which is not the case. We propose to replace the current title by 'Stochastic generation of multi-site daily precipitation focusing on extreme events'. Vague reference to hydrological applications in the introduction will be removed.

- **Classification of the precipitation models:** Both referee (comment #2.3. by referee #1 and comment #2.2. by referee #2) rightly indicated that the terminology 'multi-site models' is too vague and does not describe precisely the references given afterwards. We propose to rename this class of models by 'Statistical multi-site models'. We will provide a detailed summary of the literature for this class of models, including specific extensions of the Wilks' model and how the proposed developments differ from them.

- **Mathematical formulation:** As suggested by referee #2 (comment #1.2.), a more formal mathematical formulation of GWEX could certainly improve the presentation. This will be done in the revised manuscript.
- **Names of the models:** As indicated by referee #1 (comment #1.2.), the current model names are confusing. New names will be given to the different model versions.

- **Flowchart of the models:** As suggested by referee #1 (comment #3.2.), a flow chart could be added in order to clarify the modifications made to the original Wilks' model and to illustrate the different model versions. A flow chart will thus be added to the revised version of the manuscript.

- **Specific sections devoted to parameter estimation and simulation:** In the current version of the manuscript, the methods applied to estimate the model parameters are described all along the different sections. In the same way, details about the generation of the scenarios were provided in sections 2.3 and 3.3. As suggested by referee #2 (comments #1.2., #2.5. and #2.6.), specific sections will thus be devoted to the estimation and simulation steps.

**Reduced significance of the results without an hydrological application**

The referee #1 raised a concern about the hydrological application (comments #1.4., #2.12., #2.14., #2.16.). In particular, according to the referee, the pertinence of this study can be questioned as the relevance of some metrics (extreme precipitation amounts at different temporal and spatial scales) cannot be proven without an hydrological application. The two following paragraphs motivate the choice of these metrics and explain why the hydrological evaluation is not carried out in this study.

First, we would like to remind the key motivation of this study. The proposed stochastic models intend to preserve the most critical properties of precipitation at different spatial and temporal scales, and especially extreme precipitation amounts. We believe that a precipitation model which has these properties has a better chance to reproduce adequately flood properties for small sub-catchments as well as for large basins. Furthermore, empirical evidences have been provided by Froidevaux (2014) and Froidevaux et al. (2015) in our study area (i.e. Switzerland). Using 60 years of gridded precipitation data, Froidevaux et al. (2015) show that, in Switzerland, the generation of floods is mainly influenced by areal precipitation amounts accumulated on short periods (e.g. 1 to 3 days). Typically, the 2-day precipitation sum before floods is the most correlated to the flood frequency and the flood magnitude. These results are obtained by analyzing a wide variety of catchments, their areas ranging from 10 km$^2$ to 12,000 km$^2$. This study clearly motivates the multi-scale evaluation in space and time and the relevance of the precipitation metrics shown in our manuscript. These studies have been very briefly mentioned at the beginning of p.12 and these results must be discussed in more details, as will be done in the revised manuscript.

Second, we agree that hydrological applications would validate the importance of such properties. Actually, hydrological applications are currently undertaken by the University of Zürich. A conceptual hydrological model (HBV) is applied to 87 sub-basins partitioning the whole study area, using precipitation scenarios produced by GWEX as inputs. Numerous technical issues still need to be resolved. Some basins are ungauged, or with very short streamflow series. The hydrological system of the Aare-Rhine river needs to be treated as a whole since floods at larger spatial scales need also to be investigated. Rating curves have very high uncertainties in some basins and need to be re-evaluated. It is also important to note that this hydrological study (as well as our study) is particularly challenging considering the large spatial extent of the Aare river catchment. These studies stand out from similar studies which are usually limited to few precipitation stations and one "small" catchment (see, e.g., Keller et al., 2015, recently published in HESS, with an application to 8 precipitation stations located in a catchment with a size of 1700 km$^2$, to be compared with our study area of 17,000 km$^2$). The hydrological evaluation of our weather scenarios can thus not be carried out at the present time. It should be presented in future publications, considering the complexity of this work and the amount of results. However, we agree that the hydrological application would emphasize the significance of this study, and this point must be discussed in the manuscript.

**Validation and choice of metrics**

Both referees (comments #1.7., #1.8., #2.5. by referee #1, and comment #1.3. by referee #2) suggested additional validation criteria. Following their suggestions, QQ-plots of the marginal distributions (empirical versus fitted E-GPD or mixture of exponentials) will be provided in the revised manuscript. Additional figures will also be added in order to assess the reproduction of lagged cross-correlations and autocorrelation of precipitation.

Comment #1.4. by referee 2, as well as comments #2.5 and #2.12. by referee 1, to a lesser extent, criticize the evaluation framework and the significance of the results concerning the reproduction of extremes. In this study, validation of extreme values is mostly performed using metrics computed at all the stations and for different spatial scales (see Figures 14 and 15 of the current manuscript). In our view, it is difficult to dismiss/validate a particular method using visual inspections of the reproduction of extremes (e.g. using Gumbel plots as in Figures 10-13 of the current manuscript, or QQ-plots). In this study, Gumbel plots are mostly shown because they illustrate interesting aspects in terms of extrapolation.

Finally, in this study, we firmly support the application of the CASE framework (Bennett et al., 2017), which enables a more systematic comparison of stochastic models. A consistent way to compute the performance metrics is important in order to obtain a fair assessment of the strengths/weaknesses of the different models. For this reason, in this study, the classification proposed by Bennett et al. (2017) is not modified. This remark will be added to the revised version of the manuscript.

**References**

Bennett, B., Thyer, M., Leonard, M., Lambert, M., and Bates, B. (2017). A comprehensive and systematic evaluation framework for a parsimonious daily rainfall field model. *Journal of Hydrology*. https://doi.org/10.1016/j.jhydrol.2016.12.043, In Press.

Froidevaux, P. (2014). *Meteorological characterisation of floods in Switzerland*. PhD thesis, Geographisches Institut, University of Bern.

Froidevaux, P., Schwanbeck, J., Weingartner, R., Chevalier, C., and Martius, O. (2015). Flood triggering in Switzerland: the role of daily to monthly preceding precipitation. *Hydrology and Earth System Sciences*, 19(9):3903–3924.

Keller, D. E., Fischer, A. M., Frei, C., Liniger, M. A., Appenzeller, C., and Knutti, R. (2015). Implementation and validation of a Wilks-type multi-site daily precipitation generator over a typical Alpine river catchment. *Hydrology and Earth System Sciences*, 19(5):2163–2177.

---

## Author Response (AR1)

**Authors reply on comment of editor**

**Comment ED:** The manuscript hess-2017-226 "Stochastic generation of multi-site daily precipitation for the assessment of extreme floods in Switzerland" has received two qualified review reports. Your replies seem to account properly to all Reviewers' comments. Thus, I would like to invite you to submit a revised version of the manuscript, which will be put again at the attention of the two Reviewers.

We thank the editor for his response. Following Reviewers' comments, major changes have been made to the manuscript. This document details these changes (with the line numbers corresponding to the marked-up manuscript version, with track changes) and provides complete replies to referee #1 and #2. We hope that the revised manuscript responds to their concerns.

**Authors reply on comments of referee #1**

We thank the referee for this thorough review and for the numerous constructive suggestions. The general presentation of the manuscript has been modified following these suggestions.

**1. General comments**

**Comment R1 #1.1:** The title of the paper is a bit misleading. The three models may be used for the spatial assessment of floods and hydrological modelling is mentioned not only in the title, but also throughout the manuscript. However, the precipitation models are not applied in an impact assessment in this study and for this reason in my eyes the title should solely contain the comparison of three precipitation models. It is a bit irritating that the authors refer to the importance of several aspects of the precipitation model performance whose importance is not really demonstrated.

We agree that the title was misleading. It has been replaced by 'Stochastic generation of multi-site daily precipitation focusing on extreme events'. We think that it is important to indicate the emphasis on the reproduction of very large precipitation events, in terms of intensity, duration, and spatial extent.

**Comment R1 #1.2:** The names of the new precipitation models are a bit misleading. First, "1D" and "3D" give the impression of any type of one- and three-dimensional simulation methodology. However, they represent days ("D"). I would rename the models into something more suitable.

This is an excellent suggestion and the names for these versions have been replaced by:

1. Wilks: the model proposed by Wilks (1998),

2. Wilks_EGPD: A first direct extension of 'Wilks', with the E-GPD and a Markov chain of order 4, as suggested by the referee (see comment R1 #1.6),

3. GWEX: the current GWEX-1D model,

4. GWEX_Disag: the current GWEX-3D model. It clearly indicates the disaggregation step which follows up the simulation at a 3-day scale.

**Comment R1 #1.3:** As far as I understand from the paper, the new GWEX models are actually "Wilks models" but with a new method to simulate the precipitation amounts (using temporally and spatially correlated random numbers from an autoregressive process and using a Student copula for the spatial component). I think this should be stated as such in the paper as the manuscript presents the new models more as a revolution rather than an evolution. So one of the first sentences could be that the paper deals with two modifications of the Wilks approach.

We agree. The fact that GWEX are evolutions of the Wilks model must be clearly stated. In fact, it was already indicated at p.2/l.25 and p.4/l.10 (first version of the manuscript) and throughout the presentation of the models. As suggested by the referee, this point is now indicated directly in the abstract and in the introduction (see p.1/l.3 and p.3/l.15). However, it must be underlined that GWEX is a significant evolution of the model introduced by Wilks (1998). First, as indicated by the referee, the methodology applied to simulate the precipitation amounts is considerably modified. We consider different temporal and spatial dependences, and we also discuss the choice of the marginal distribution in details, which is currently overlooked in the literature of precipitation stochastic models. Second, GWEX-3D (which will be named GWEX_Disag) combines simulations at a 3-day scale and a dis-aggregation approach, which represents a further step in the complexity of

the model. In our opinion, GWEX cannot only be considered as a slight modification/evolution of the Wilks model.

**Comment R1 #1.4:** The motivation behind the study and for the new model developments is the impact assessment. However, without the same the reader will not be able to really understand the sensitivity of certain statistics in regard to the assessment of extreme floods. I think the importance of some of the statistical metrics should be explained in more detail referring to the area of their application, and proof must be given of their relevance. Other literature in such a study (complex mountain region) is not very convincing to me.

We thank the referee for this suggestion and additional details regarding the importance of the statistical metrics have been provided in the revised version (section 2, p.5, lines 2-4; section 4, p.16, lines 4-9). In particular, Froidevaux (2014) analyze meteorological events triggering floods in Switzerland. These studies were very briefly mentioned at the beginning of p.12 in the original manuscript and these results are now discussed in more details. However providing a proof of the relevance of these metrics seems complicated without the hydrological application (which is clearly beyond the scope of this paper, as discussed in comment R1 #2.14.). If the referee has more specific metrics that could be presented, we would be glad to include them in our study.

**Comment R1 #1.5:** The abstract is incomplete and must be much more detailed and specific. What is an "event"? What is "large"? What are "recent advances"? The abstract should mention the Wilks model, the two new models (maybe also a short sentence how they work) and the basic outcomes of the study.

We thank the referee for this constructive suggestion. The abstract has been substantially modified and extended.

**Comment R1 #1.6:** The Wilks model could likewise be applied with E-GPD distributions for precipitation intensities and Markov chains of the order 4. That is, revealed weaknesses of the Wilks model can easily be addressed. I recommend adapting the Wilks approach for a more objective comparison. The original Wilks approach is not a given and was just one application for a specific dataset in the US and in my eyes it should always be revised for other study areas and climates.

We thank the referee for this suggestion. An additional version of the Wilks model, Wilks_EGPD, has been added to the comparison of the three previous models. Wilks_EGPD considers a E-GPD instead of a mixture of exponential distributions for the marginal distributions, and a Markov chain of order 4 instead of order 1.

**Comment R1 #1.7:** For flood modelling, the lagged cross correlations (see Wilks 1998, page 183) can be very important as they represent the progression of weather systems across the study area. Especially at larger scales the progression of weather events may be important. I strongly recommend plotting these statistics for all three models.

We thank the referee for this recommendation, which is also proposed by the referee #2. Lagged and unlagged inter-site correlations of precipitation amounts are now presented in Figure 10 (p.26) in Section 5.2.

**Comment R1 #1.8:** The autocorrelation of precipitation is addressed by MAR(1) models in the GWEX models. I would recommend plots for the autocorrelation of the precipitation intensities for some sites to see potential differences in their performance.

We thank the referee for this suggestion. Figure 10 (p.26) also presents an assessment of the autocorrelation of the precipitation amounts at the stations (black points), together with the cross-correlations (gray points).

**2. Specific comments**

**Comment R1 #2.1.** Line 8. I think there is a language issue.

This sentence has been reformulated in the revised version (see p.1, lines 14-16).

**Comment R1 #2.2.** Line 10. Not only conceptual models. There are more recent studies for coupling WGs with impact models.

Thanks for this remark. The authors are not aware of such impact models. If the reviewer has specific references, they can be included in the final version of the manuscript. The word 'conceptual' has been removed in order to include other types of hydrological models (e.g. distributed models).

**Comment R1 #2.3.** Page 1 bottom/ Page 2 top: In my eyes the classification is not fully correct. All these models are multi-site models. Also resampling methods are multi-site models. I recommend a more suitable classification even though I admit that the variety of the existing multi-site models makes a clear classification more and more difficult (also the authors combined parametric and non-parametric techniques).

We agree that the terminology 'multi-site models' is too vague and does not describe precisely the references given afterwards. This class of models has been renamed by 'Statistical multi-site models'. A detailed summary of the literature for this class of models is now provided, including specific extensions of Wilks model and how the proposed developments differ from them (p.2, lines 21-34).

**Comment R1 #2.4.** Page 4, Line 8. Are Thiessen polygons suitable for such a complex mountainous study region?

The computation of areal precipitation values is a difficult task considering the spatial and temporal variability of precipitation events, the complex topography of the study area, and the limited number of pluviographs. In Switzerland, Schäppi (2013) shows that the topography impacts rainfall amounts differently according to the type of meteorological event. In a preliminary study, the impact of different interpolation methods (inverse distance, ordinary kriging, kriging with external drift, Thiessen polygons) and different sets of stations (399, 211, 129, 47 and 22 stations) on extreme areal precipitation amounts has been analyzed. The main conclusion was that the number of stations was a much more important factor than the interpolation method. This was the main motivation for the application of the stochastic models to a high number (105) of stations. Furthermore, it is important to notice that applying more complex interpolation methods (e.g. kriging methods) increase significantly the computational cost, which can be prohibitive for the production of long meteorological scenarios.

**Comment R1 #2.5.** Page 7 "Marginal distributions". Can any proof be given that the more complex fitting of a combined distribution is really significantly better for the simulation of the extremes in this region? Also here, the most prominent argument is other literature.

QQ-plots are provided in the revised version in order to assess the quality of the fitting of these marginal distributions (p.22, Fig.6). However, it is very important to note that local applications give limited proof regarding the

performance of a distribution for the fitting of extreme values. As indicated in "Papalexiou, S. M. and Koutsoyiannis, D. (2013) Battle of extreme value distributions: A global survey on extreme daily rainfall, Water Resources Research, 49, 187201", most studies of extreme rainfall are inconclusive because they are too specific to particular areas or stations. The main explanation for these failures is that fitting and inferring the distribution tails is subject to high uncertainties in the estimation of the parameters, even for long time series (this point is also discussed and illustrated in Evin et al., 2016). The references given in the paper (Papalexiou and Koutsoyiannis, 2013; Serinaldi and Kilsby, 2014) are conclusive precisely because they are the result of a very large number of applications, and give strong arguments in favor of the application of heavy-tailed distributions. Figures 2 and 3 tend to show that low tail-distributions (like a mixture of exponentials) could lead to an under-estimation of extreme precipitations in some regions (regions where $\xi$ is different from 0, in green, yellow and red). In our study area, we acknowledge that the E-GPD does not bring a significant improvement of the performance compared to the mixture of exponential distributions (see Figure 12). However, as stated in the conclusion (p.33, lines 18-22), with only three parameters, the E-GPD provides a parsimonious and flexible representation of the whole of precipitation amounts. Its GPD tail is in agreement with recent results showing that extreme precipitation amounts must be modeled by heavy-tailed distributions (Papalexiou and Koutsoyiannis, 2013; Serinaldi and Kilsby, 2014). The general framework proposed in this paper can be applied to very distinct precipitation regimes and the possible heavy tail of the E-GPD might be valuable in other areas.

**Comment R1 #2.6.** Page 9, top of the page. If the Gaussian copula is not suitable for simulating spatially dependent extremes but the Student copula is, this could be demonstrated. I am thinking of readers who want to build the code but are not experts in copulas and want to understand the significance.

The revised manuscript includes an additional version of the Wilks model, Wilks_EGPD, with applies a E-GPD instead of a mixture of exponential distributions for the marginal distributions (see comment R1 #1.6.). The difference between Wilks_EGPD and GWEX models provides a comparison of Gaussian and Student copulas concerning the reproduction of daily precipitation extremes (see Section 5.4., p.28).

**Comment R1 #2.7.** Page 9 bottom. Why are Markov chains of the order 4 used? Have there been statistical tests or sensitivity studies to underline this decision? Later on, some remarks are given on the simulation of short dry spells, but I think this should be addressed in a more structured way.

At p.5/l.10, it was indicated that Srikanthan and Pegram (2009) apply a 4-order Markov chain and show that it improves the reproduction of dry/wet period lengths. This point is now reminded at p.14, l.8-10.

**Comment R1 #2.8.** Page 11, Table 11 (and figures). Red and green are not suitable for figures, please change the colours as some people cannot read them otherwise (https://www.nature.com/nature/journal/v510/n7505/full/510340e.html).

We thank the referee for this comment. These colors have been modified and should be suitable for most color-blind people (following the recommendations given in `https://www.nature.com/nmeth/journal/v8/n6/full/nmeth.1618.html`). As we understand this issue, it seems that types of green (bluish green) and red (vermilion) are more adapted to color-blind individuals.

**Comment R1 #2.9.** Page 12, Line 28. I guess it is very difficult to say if an extreme precipitation amount is unrealistic or not as long as they are physically possible?

It is true that an extreme precipitation amount cannot be considered as unrealistic if the amount is physically possible. However, it is difficult to define what amount can be considered as impossible. Since this constraint was not used in our applications (we always obtain $\xi < 0.25$ in our study area, see dark red areas in Fig. 2 and 3), this remark was removed from the manuscript.

**Comment R1 #2.10.** Page 16, Line 18-20. If the order of the Markov chain is the issue for short dry spells, this can be easily adapted by using the same order in the original Wilks approach. What was the argument for using the first order Markov chains in the Wilks model? (see comment above)

As indicated in p.14/l.8-10, the direct extension of the Wilks model, Wilks_EGPD, is used to illustrate the impact of using a Markov chain of order 4 compared to order 1.

**Comment R1 #2.11.** Page 21, Line 8-9. Please explain the seasonal differences with explicit reference to the study area and its climatology for

better understanding.

Considering the number of additional figures provided in the revised manuscript, and considering that the assessment of the inter-annual variability is not central in this study (see Comment R1 #2.17.), this plot has been removed from the revised version of the manuscript.

**Comment R1 #2.12.** Page 22, section 4.4. and figure 10. To me, the performance looks fair for all three models. The main difference is the simulation of higher extremes with the GWEX models. The authors mention the difference but it needs further discussion. Also, how can we know that the extremes of one method are more realistic than from another? While we know little about the validity of the simulated extremes, they may have a big impact on simulated floods, especially in small catchments (but as mentioned before, this is not examined in the paper).

We agree with the referee, the performance looks fair for all three models if we look at figure 10 of the original manuscript. However, this figure only points out differences of behavior between the three models. As mentioned above (see Comment R1 #2.5.), these illustrative examples cannot be used to test the performance of the different models in regard to extreme daily precipitation amounts. The only way to perform such a validation is to apply some metrics on a large set of applications (here, for example, at all the stations), which is done at Figures 12 and 13 of the revised manuscript. Figures 10-13 of the original version, which were showing the fitting of the annual maxima at some stations/basins, have been removed from the manuscript.

**Comment R1 #2.13.** Page 26 Line 10-13. It is not surprising that the non-parametric disaggregation leads to a better performance. I understand its strengths but it may likewise be a limiting factor in generating extremes. In our opinion, GWEX_Disag represents a compromise between a purely statistical approach and a nonparametric approach. In terms of possible simulated amounts, it is not limiting factor at a 3-day scale and only is a constraint concerning the repartition of 3-day amounts across the daily steps.

**Comment R1 #2.14.** Page 29, first line 2-9. As already mentioned, I see the motivation behind the study (and it is generally a good one). But without any proof that the differences in the performance of the three precipitation models really have a significant impact on the simulation results of hydrological extremes (also considering all the uncertainties in hydrological

models), the significance of the research outcomes remain questionable.

We appreciate this criticism. The two following paragraphs motivate the assessment of these extreme precipitation amounts at different temporal and spatial scales and explain why the hydrological evaluation is not carried out in this study.

First, we would like to remind the key motivation of this study. The proposed stochastic models intend to preserve the most critical properties of precipitation at different spatial and temporal scales, and especially extreme precipitation amounts. We believe that a precipitation model which has these properties has a better chance to reproduce adequately flood properties for small sub-catchments as well as for large basins. Furthermore, empirical evidences have been provided by Froidevaux (2014) and Froidevaux et al. (2015) in our study area (i.e. Switzerland). Using 60 years of gridded precipitation data, Froidevaux et al. (2015) show that, in Switzerland, the generation of floods is mainly influenced by areal precipitation amounts accumulated on short periods (e.g. 1 to 3 days). Typically, the 2-day precipitation sum before floods is the most correlated to the flood frequency and the flood magnitude. These results are obtained by analyzing a wide variety of catchments, their areas ranging from 10 km$^2$ to 12,000 km$^2$. This study clearly motivates the multi-scale evaluation in space and time and the relevance of the precipitation metrics shown in our manuscript. These studies were very briefly mentioned at the beginning of p.12 and these results are now discussed in more details in the revised manuscript (see section 2, p.4, lines 2-4; section 4, p.16, lines 1-9).

Second, we agree that hydrological applications would validate the importance of such properties. Actually, hydrological applications are currently undertaken by the University of Zürich. A conceptual hydrological model (HBV) is applied to 87 sub-basins partitioning the whole study area, using precipitation scenarios produced by GWEX as inputs. Numerous technical issues still need to be resolved. Some basins are ungauged, or with very short streamflow series. The hydrological system of the Aare-Rhine river needs to be treated as a whole since floods at larger spatial scales need also to be investigated. Rating curves have very high uncertainties in some basins and need to be re-evaluated. It is also important to note that this hydrological study (as well as our study) is particularly challenging considering the large spatial extent of the Aare river catchment. These studies stand out from similar studies which are usually limited to few precipitation stations and one "small" catchment (see, e.g., Keller et al., 2015, recently published in

HESS, with an application to 8 precipitation stations located in a catchment with a size of 1700 km$^2$, to be compared with our study area of 17,000 km$^2$). The hydrological evaluation of our weather scenarios can thus not be carried out at the present time. It should be presented in future publications, considering the complexity of this work and the amount of results. However, we agree that the hydrological application would emphasize the significance of this study, and this point is discussed in the last section (Section 6, end of page 33, top of page 34).

**Comment R1 #2.15.** Page 29, Line 21-22. Please explain why, see comments above.
*See Comment R1 # 2.11.*

**Comment R1 #2.16.** Page 29, Line 27-28. The issue of larger spatial scales could be addressed by running more analyses at smaller scales. So the key motivation of the study is probably to examine large flood events and their spatial dependences? If so, this should be better explained. But again, without really simulating the floods throughout different scales the arguments for a particular precipitation model choice is questionable.
The key motivation is to develop a stochastic model for precipitation which preserve the most critical properties of precipitation at different spatial and temporal scales, and especially for extreme precipitation amounts. A meteorological model that does not preserve these metrics is unlikely to reproduce adequately flood properties for small sub-catchments as well as for large basins. However, we agree that the assessment of large flood events, in particular their spatial dependency, is very important. This will be done in further studies (see comment R1 #2.14.) by other research teams involved in this project.

**Comment R1 #2.17.** Page 30. Is the underestimation of the inter-annual variability such a big issue in Switzerland and for flood modelling? I would assume it is more an issue in more arid regions and for example agricultural studies? Some more remarks on the relevance in Switzerland and floods in general would be useful.
Thanks for this remark. We agree that the inter-annual variability is not central in this study, considering that we are interested in flood risk assessment. Indeed, this issue is more critical for other hydrometeorological applications, including agricultural and water resource related ones. Consequently, as indicated in comment R1 # 2.11., this analysis has been removed from the manuscript.

**3. Summary of review**

**Comment R1 #3.1.** The abstract needs revision and must be more detailed (see general comments).
See comment R1 # 1.5.

**Comment R1 #3.2.** The introduction is not very well structured. The arguments for the construction of the new precipitation methodologies are mainly based on other literature and reasoning. The context of the paper should (i) either be revised (comparison of precipitation models) or (ii) proof must be given of the advantages using the new models by really coupling them with a hydrological model and examining the estimated flood events in the study region. I think it is the key weak point of the paper: reference is given to an application, which is not really done. Also, the title and abstract are a bit misleading and the reader may expect a flood modelling study and thus more than what has been presented.
We agree that the introduction was misleading and it has been modified in order to clearly indicate that this study aims at comparing precipitation models, the hydrological context being the motivation for the thorough assessment of areal precipitation extremes.

**Comment R1 #3.3.** For the three different precipitation models, I would recommend a flow chart with the Wilks model as the central component and then the adaptations that have been done. This makes it easier for the reader to understand all models and what has been changed.
This is an excellent suggestion and a flow chart has been added to the revised version of the manuscript (see Figure 3).

**Comment R1 #3.4.** Although the level of English is very good, some (minor) mistakes can be found in the manuscript and a native speaker should probably have a final look before resubmission.
A professional native English editor has been hired to proofread the final version of the revised manuscript.

**Authors reply on comments of referee #2**

**Summary**

The authors propose extensions of a classical multisite daily rainfall generator initially proposed by Wilks in 1998. The framework of Wilks model is flexible enough to allow many adaptations, and the authors of this paper propose - to add more structure in the dynamics of the model by considering higher order Markov model for the occurrence process and an autoregressive component for the amounts - to use a hybrid distribution for the marginal distribution to deal with heavy tail distributions - to use a Student copula for the spatial structure to catch upper tail dependence. I believe that all these extensions make sense and are interesting to try.
We thank the referee for this review and for these constructive comments. Most of the following suggestions have been incorporated in the modified manuscript.

**1. General comments**

**Comment R1 #1.1.** Many extensions of the Wilks model have already been proposed in the literature. I think that a review of this literature must be included in the paper and that the authors should explain why the extension that they propose is original and useful with respect to this literature.
We agree that the differences between GWEX and the existing extensions of Wilks model must be presented in the introduction. A more complete presentation of the literature has been included in the introduction (see p.2, lines 26-34) of the revised version of the manuscript.

**Comment R1 #1.2.** In my opinion, one weakness of the paper is that the model is formulated as a simulation tool rather than as a proper statistical model. It is also the case for the original Wilks model, but it has then been reformulated by other authors as a statistical model, see e.g. Thompson et al. (2007). I think that the paper would be easier to read for statisticians like me if a similar formalization was done in the paper. In particular, the

various assumptions on the occurrence/amount processes should be written precisely using formulas and the definition of the model should be separated from the discussion on parameter estimation and simulation.

We thank the reviewer for this excellent suggestion. GWEX is now presented using a more formal mathematical formulation and the whole section 3. ("Multi-site precipitation model") has been modified. A specific section is now devoted to parameter estimation (section 3.3, p.11-13).

**Comment R1 #1.3.** I believe that the validation part must also be improved. First, some usual validation criteria for rainfall generators, such as diagnostics based on the marginal distribution (e.g. qqplot) and the second order structure of the process (autocorrelation and crosscorrelation functions) are not shown and it makes it difficult to see the benefit of using a hybrid distribution and the autoregressive component. Also the chosen validation criteria does not permit to see the interest of using a student Copula (does it really improve the modeling of extremal dependence?).

These remarks have also been made by the referee #1 (comments R1 #1.8, R1 #2.5 and R1 #2.6). QQ-plots are now provided to assess (visually) the quality of the fitting for the marginal distributions (see Figure 6, p.23). An additional figure provides an assessment of the performance concerning the autocorrelations and the reproduction of cross-correlations (see Figure 10, p.26). Finally, an additional model version, a direct extension of Wilks model, with the E-GPD instead of a mixture of exponential distibutions, "Wilks_EGPD", has been added to the three original models. This additional version enables the assessment of the impact of the Student copula (versus a Gaussian spatial structure).

**Comment R1 #1.4.** Finally, I find the simulation results generally disappointing. If I understand correctly the categorization, we should obtain about 90% of good if the model was able to reproduce the statistics of the observed rainfall? Is it satisfactory to obtain percentage around 50%?.

Yes, we should obtain about 90% of good if the model is able to reproduce the observed statistics, and very few 'poor' cases. As indicated in the paper, our primary criteria to judge the overall performance of a model is the number of metrics for which 'poor' performances are obtained. We agree that these percentages are subjective (why 90%? Is 50% of good cases good enough?) but not more subjective, in our opinion, that the visual inspection of a QQ-plot. Furthermore, the purpose of the CASE framework, as

presented in Bennett et al. (2017), is to enable a more systematic comparison of stochastic models. Our study also tries to promote this approach. A more systematic comparison of the models, which includes a consistent way to compute the performance metrics, is important in order to obtain a fair assessment of the strengths/weaknesses of the different models. For this reason, this study applies the classification proposed by Bennett et al. (2017), without modifying the classification.

**2. Specific comments**

**Comment R1 #2.1.** Keywords are missing?
In HESS, to the extent of our knowledge, keywords do not appear in the manuscript.

**Comment R1 #2.2.** End of Page 1/top of page 2. I am not really satisfied by the proposed classification. For example weather type models are often used as multisite rainfall generators (without conditioning to large scale information). Also it would be useful to cite the review papers on rainfall generators here.
We agree that the terminology 'Multi-site models' is too vague here. A similar comment has been done by the referee #1 (see comment R1 #2.3.). We propose to replace 'multi-site models' by 'statistical multi-site models' (see p.2, lines 21-34). Additional references have been incorporated.

**Comment R1 #2.3.** Section 2.1. The authors go directly from a Markov chain of order p=1 to a Markov chain of order p=4. I would expect that the best value of $p$ is somewhere between these two values. The authors could try to find the optimal value of $p$, using for example standard model selection criteria.
We thank the reviewer for this suggestion. It is true that an optimal value might be found if there was an easy selection criteria. As this point is not central in our study, a direct comparison of Markov chains of order $p = 1$ and $p = 4$ is deemed sufficient.

**Comment R1 #2.4.** Equation (5). I am surprised that the authors use a diagonal matrix for $A$. I would expect that it is useful to add some spatial structure here?
Initial versions of GWEX were applying a full covariance matrix for $A$. However, it seems that large covariance matrices are often very close to a non positive definite matrix. This is not really problematic during the estimation step, but leads to very unstable results during the simulation step. As applying a diagonal matrix for $A$ does not seem to degrade the performance of GWEX, this solution was retained.

**Comment R1 #2.5.** Section 2.3 and 3.3 should be merged.
We thank the reviewer for this suggestion. Following comment R2 #1.2., these sections have been re-organized with a specific section devoted to the estimation step. Previous section 3.3 has been removed in the revised manuscript.

**Comment R1 #2.6.** Section 3. Why is it called "Application"? I do not see any application here.
Following previous comments (comments R2 #1.2. and R2 #2.5.), the whole section 3 has been reorganized. Section 3.1 'Split-sampling procedure' is now presented at the beginning of section 5 "Results" (see p.19). Previous section 3.2 'Regionalization of the $\xi$ parameter' is related to the estimation of the parameters is now presented in section 3.3, p.11-13. Previous section 3.3 'Generation of scenarios' has been removed (see previous comment).

**Summary of changes**

**Overall presentation of the manuscript**

Most of the referee's comment are related to the presentation of the methodology and the results. These comments are entirely justified and are appreciated, as they greatly enhance the paper. The following paragraphs summarize what modifications have been made to the manuscript (more details can be found in the response to specific comments):

- **Abstract:** We agree with the referee #1 (comments #1.5. and #3.1.) that the abstract was not specific enough. Additional details have been added (summary of the model developments, key results, etc.)

- **Title and introduction misleading:** As pointed out by the referee #1 (comments R1 #1.1. and R1 #3.2.), the title and the introduction

seemed to indicate that our study shows the results of an hydrological application, which is not the case. The title has been replaced by 'Stochastic generation of multi-site daily precipitation focusing on extreme events'. Vague references to hydrological applications in the introduction have been been removed.

- **Classification of the precipitation models:** Both referees (comment R1 #2.3. and comment R2 #2.2.) rightly indicated that the terminology 'multi-site models' was too vague and did not describe precisely the references given afterwards. This class of models is now named 'Statistical multi-site models'. A detailed summary of the literature for this class of models is provided, including specific extensions of Wilks model and how the proposed developments differ from them.

- **Mathematical formulation:** As suggested by referee #2 (comment R2 #1.2.), we now present a more formal mathematical formulation of GWEX (section 3).

- **Names of the models:** As indicated by referee #1 (comment R1 #1.2.), the current model names are confusing. New names have been given to the different model versions.

- **Flowchart of the models:** As suggested by referee #1 (comment R1 #3.2.), a flow chart has been added in order to clarify the modifications made to the original Wilks model and to illustrate the different model versions (see Figure 6).

- **Specific section devoted to the parameter estimation:** As suggested by referee #2 (comments R2 #1.2., R2 #2.5. and R2 #2.6.), a specific section is now devoted to the estimation step.

**Validation and choice of metrics**

Both referees (comments R1 #1.7., R1 #1.8., R1 #2.5. and R2 #1.3.) suggested additional validation criteria. Following their suggestions, QQ-plots of the marginal distributions (empirical versus fitted E-GPD or mixture of exponential distributions) is now presented in the revised manuscript (see Figure 6). Additional figures have also been added in order to assess the reproduction of lagged and unlagged cross-correlations (see Figure 10).

Comment R2 #1.4., as well as comments R1 #2.5 and R1 #2.12., to a lesser extent, criticize the evaluation framework and the significance of the results concerning the reproduction of extremes. In this study, validation of extreme values is mostly performed using metrics computed at all the stations and for different spatial scales. In our view, it is difficult to dismiss/validate a particular method using visual inspections of the reproduction of extremes (e.g. using Gumbel plots as in Figures 10-13 of the original manuscript, or QQ-plots). Consequently, previous Figures 10-13 have been removed from the manuscript. These figures were mostly shown to illustrate interesting aspects in terms of extrapolation but seem to be prone to different interpretations in terms of performance. Finally, we now present relative differences in Figures 12-13 (instead f absolute differences), in order to highlight potential under/overestimations at large spatial scales.

In this study, we firmly support the application of the CASE framework (Bennett et al., 2017), which enables a more systematic comparison of stochastic models. A consistent way to compute the performance metrics is important in order to obtain a fair assessment of the strengths/weaknesses of the different models. For this reason, in this study, the classification proposed by Bennett et al. (2017) is not modified. A remark has been added to the revised version of the manuscript (p.16, lines 16-18).

**Parameter estimation of the inter-site correlations, and autocorrelations**

As indicated in Wilks (1998), direct estimates of the spatial and temporal dependence of precipitation amounts cannot be obtained since non-zero precipitation amounts $Y_t(k)$ is a hidden variable which cannot be observed. In the previous version of the manuscript, these correlations were directly estimated from positive precipitation amounts. However, this method leads to a significant under-estimation of the inter-site correlations of precipitation amounts (zero and non-zero). In the revised version of the manuscript, we follow the methodology proposed by Wilks (1998) and Keller et al. (2015). For each pair of stations, we generate long sequences of precipitation amounts $P_t(k)$ using the estimated parameters of the occurrence process ($\hat{\Pi}$ and $\hat{\omega}_{kl}$), the parameters of the marginal distributions and a correlation coefficient $m_0(k, l)$ indicating the degree of spatial dependence. Similarly to the occurrence process, $\hat{m}_0(k, l)$ is then found iteratively by matching the correlation between

these long random streams with the observed correlation $\mathrm{Corr}(P_t(k), P_t(l))$ (see Wilks, 1998; Keller et al., 2015, for further details). The correlation matrix $\hat{\mathbf{M}}_0$ is then composed of the cross-correlations $\hat{m}_0(k, l)$ obtained for all possible pairs of stations. For each station, the estimates of the lag-1 serial correlation coefficients of the matrix $\mathbf{A}$ are obtained using the same simulation approach (see end of page 12, top of page 13).

These modifications improves greatly the reproduction of extreme precipitation amounts at large spatial scales, in particular for model GWEX concerning 3-day precipitation extremes.

[revised manuscript text omitted]